# TAMPO: Task- and Model-Aware Automatic Prompt Optimization for Auto-Routing in LLM-based Systems

Yuejun Jiao [1]  Jun Xia [1,2]  Yanxin Yang [1]  Yonghao Yang [1]  Hao Shen [1]  Mingsong Chen [1]

## Abstract

Automatic Prompt Optimization (APO) enables Large Language Models (LLMs) to adapt to specific tasks while minimizing manual engineering costs. However, since existing APO approaches either rely solely on multi-round iterative procedures or use model-specific generators tailored to optimizing prompts for a single model and objective, they are not readily applicable to auto-routing scenarios, which require operating over diverse LLMs and juggling multiple, often competing, trade-offs. To address this issue, we propose TAMPO, a novel task- and model-aware APO framework for auto-routing in LLM-based systems. Specifically, to capture performance variation across a broad range of tasks and models, we construct a comprehensive, heterogeneity-aware dataset to train an uncertainty-aware reward model. Serving as an offline proxy, this reward model can greatly mitigate reward hacking, allowing TAMPO to learn an optimal multi-objective conditional policy for robust prompt generation. Based on the user requirements encoded in our defined preference vector, this policy enables flexible control over prompt generation and supports a cost-effective deployment strategy. Extensive experiments across 86 tasks demonstrate that TAMPO effectively maintains performance stability across diverse tasks and models, providing a robust, controllable solution for auto-routing in various LLM-based systems.

## 1. Introduction

Large Language Models (LLMs) exhibit strong performance on a variety of natural language processing tasks, such as reasoning, summarization, and knowledge-intensive question answering (OpenAI, 2023; Touvron et al., 2023). However, their effectiveness is highly dependent on prompt design, which critically shapes task interpretation, reasoning trajectories, and final outputs (Wei et al., 2022). Even minor variations in prompt phrasing can cause pronounced changes in response quality, latency, and token usage (Zhao et al., 2021; Lu et al., 2022). This sensitivity motivates Automatic Prompt Optimization (APO), which aims to reduce the effort of manual prompt engineering and promote stable model behavior across tasks (Zhou et al., 2023; Yang et al., 2024). Modern LLM-based systems are increasingly leveraging auto-routing pipelines (Chen et al., 2024; Ong et al., 2024), in which prompt selection becomes a dynamic routing decision rather than a choice among static templates. In such settings, a prompt must simultaneously i) match the task semantics, ii) be compatible with model-specific properties (e.g., architectures and quantization configurations), and iii) meet cost-effective deployment requirements (e.g., latency). These combined demands impose tight efficiency constraints on prompt optimization.

Unfortunately, existing APO approaches are not designed to jointly address these requirements, as they are primarily aimed at single-objective optimization. Specifically, online APO methods (Zhou et al., 2023; Yang et al., 2024; Fernando et al., 2024) depend on performing multiple refinement rounds to enhance response quality, which leads to substantial inference delays. In contrast, offline APO methods (Deng et al., 2022; Diao et al., 2023) are tailored to specific models, as they require training an individual prompt generator for each model, thereby lacking general, model-aware adaptability. Moreover, most APO approaches optimize for a single objective (i.e., response quality) while overlooking user-centric factors (e.g., inference latency and token usage), which are crucial for decision-making in auto-routing settings. *Therefore, a key challenge in APO is how to rapidly generate high-quality prompts for auto-routing that fully leverage the strengths of different LLMs, while jointly satisfying the users' preferred multi-objective constraints.*

To address this challenge, we propose TAMPO, a novel Task-And Model-aware automatic Prompt Optimization framework that formulates prompt generation as a conditional

---

[1]East China Normal University [2]Hong Kong University of Science and Technology. Correspondence to: Mingsong Chen <mschen@sei.ecnu.edu.cn>.

*Proceedings of the 43rd International Conference on Machine Learning*, Seoul, South Korea. PMLR 306, 2026. Copyright 2026 by the author(s).

learning task, jointly leveraging both task properties and model-specific metadata. The framework begins by building a prompt performance dataset through systematic evaluation over diverse tasks and model setups. An uncertainty-aware reward model is then trained to estimate response quality, inference latency, token usage, and corresponding uncertainty. Using the reward model for guidance, a Reinforcement Learning (RL)-based prompt-generation policy produces prompts that negotiate trade-offs among multiple objectives across different tasks and model configurations. By moving prompt optimization from online search to offline learning, TAMPO delivers a robust, controllable mechanism for APO tailored for auto-routing scenarios. This paper makes the following four major contributions:

1. We cast prompt optimization as a conditional generation task that explicitly incorporates model metadata, aiming to mitigate severe performance drops due to model heterogeneity and thereby maintain robustness across a wide range of deployment settings.
2. We build a comprehensive, heterogeneity-aware dataset to measure prompt performance across diverse tasks, model architectures, and quantization settings, aiming to capture relationships between prompt patterns and evaluation metrics.
3. We propose an RL-based approach that directly optimizes the Pareto frontier over quality, latency, and token usage, enabling prompt strategies to be dynamically adjusted in response to real-time constraints.
4. We introduce an uncertainty-aware reward model that jointly predicts performance metrics and their associated uncertainty, enabling more reliable performance ranking and reducing the risk of reward hacking.

## 2. Problem Formulation

### 2.1. Formal Problem Definition

We formalize APO as a conditional generation problem (Deng et al., 2022). Formally, we seek to train a policy $\pi$ that maps the joint space of task information $\mathcal{T}$ and model metadata $\mathcal{M}$ to the prompt space $\mathcal{P}$. Specifically, this involves optimizing a conditional policy $\pi$ that dynamically generates a prompt $p = \pi(\cdot|t, m)$ adapted to the constraints of a task $t \in \mathcal{T}$ and a target model $m \in \mathcal{M}$. Meanwhile, we model the prompt evaluation process $\mathcal{E}$ as a stochastic process governed by the environment dynamics. This process captures both the LLM's intrinsic variability in generation and performance fluctuations caused by system-level factors (Wang et al., 2023). Given the multi-objective nature of the environment, we define the objective set $\mathcal{K} = \{\text{qual}, \text{lat}, \text{tok}\}$ corresponding to response quality, latency, and token consumption. The evaluation result $Y$ is sampled from this evaluation function:

$$Y = (y_k)_{k \in \mathcal{K}} \sim \mathcal{E}(t, m, p). \tag{1}$$

### 2.2. Offline Optimization Framework

A critical constraint in auto-routing scenarios is the high computational overhead of model switching, which prevents repeated online interactions (Levine et al., 2020). Therefore, directly maximizing the expectation of $\mathcal{E}$ via trial-and-error online is intractable, necessitating an offline approach. To address this challenge, we decompose the problem into two subproblems offline: i) uncertainty-aware reward model training and ii) multi-objective conditional policy optimization. Specifically, we train the uncertainty-aware reward model $R_\theta(t, m, p)$ as an offline proxy using a heterogeneity-aware dataset comprising $(t, m, p)$ triplets and their corresponding performance metrics. In particular, since the policy inevitably explores novel prompts beyond the coverage of any static dataset (Yu et al., 2020; Kidambi et al., 2020), the reward model must quantify epistemic uncertainty in performance to prevent reward hacking from over-optimistic predictions in these unseen regions. To satisfy user-specified requirements, we introduce a preference vector $\mathbf{w} \in \mathbb{R}^{|\mathcal{K}|}$, which assigns weights to each objective $k \in \mathcal{K}$ to support controllable prompt generation. The multi-objective conditional policy $\pi_\phi$ generates prompts conditioned on the task-model pair $(t, m)$ and a preference vector $\mathbf{w}$ over multiple objectives:

$$p \sim \pi_\phi(\cdot|t, m, \mathbf{w}). \tag{2}$$

Through this generation process, the policy approximates the Pareto frontier by ensuring that the generated prompts achieve the trade-offs defined by $\mathbf{w}$ in a controllable, task- and model–aware manner in auto-routing scenarios.

## 3. Methodology

### 3.1. Overview of TAMPO

Figure 1 illustrates the framework and workflow of TAMPO. In our approach, we leverage various iterative strategies to construct a heterogeneity-aware dataset that links prompt candidates to their multi-objective performance metrics across varied task and model configurations. We use the dataset to train an uncertainty-aware reward model that serves as a reliable proxy for evaluating quality, latency, and token consumption. Guided by this proxy, the multi-objective conditional policy is trained through a two-stage process that enhances stability and controllability. Specifically, we first employ Supervised Fine-Tuning (SFT) (Ouyang et al., 2022) on high-quality samples to establish a stable initialization, and then proceed to an RL stage using Group Relative Policy Optimization (GRPO) (Shao et al., 2024) with Dirichlet weight sampling to explicitly explore the Pareto frontier. During inference, the policy acts as a conditional generator, producing prompts that simultaneously satisfy specific task and model constraints and preference requirements.

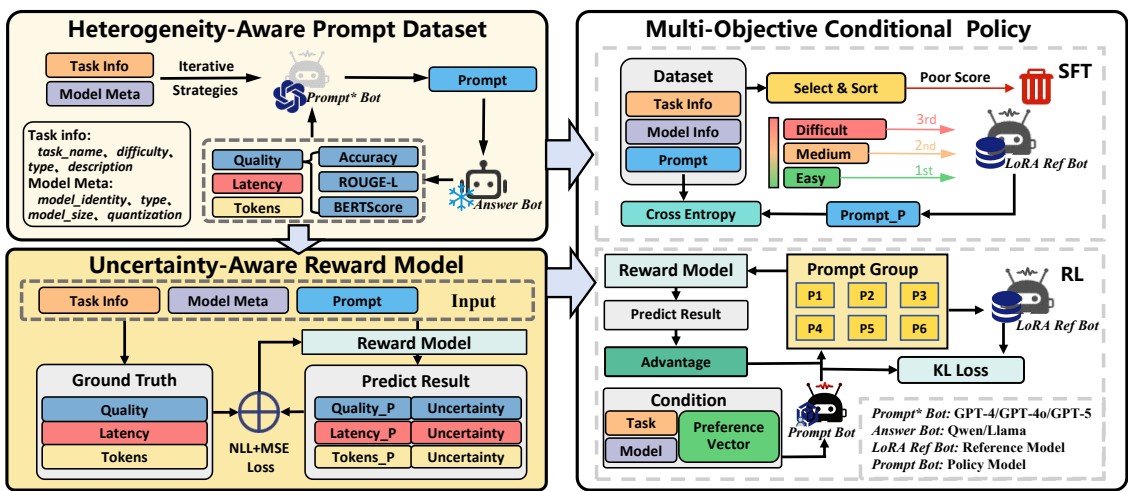

*Figure 1.* The framework and workflow of TAMPO.

### 3.2. Heterogeneity-Aware Dataset

To support task- and model-aware prompt generation under auto-routing inference, we construct a heterogeneity-aware dataset that captures prompt behaviors across diverse task and model configurations. Each entry aggregates task information, model metadata, candidate prompts, and the multi-objective metrics. Specifically, task information includes the task name, type, description, and difficulty, while model metadata encodes deployment-relevant attributes such as model identity, parameter scale, and quantization level. By evaluating candidates under these varying conditions, we record response quality (e.g., Accuracy or ROUGE-L), latency, and token consumption to establish a robust mapping between prompt patterns and inference outcomes.

We generate these prompt candidates in the dataset through a multi-round optimization process that integrates four key iterative strategies: i) to ensure **style diversity**, we utilize distinct high-capability LLMs (e.g., GPT-4, GPT-4o and GPT-5) as prompt generators to induce diverse prompting styles, ranging from few-shot to Chain-of-Thought (CoT) (Wei et al., 2022; Kojima et al., 2022); ii) for the **elimination mechanism**, to retain prompt candidates exhibiting varying performance levels for contrastive learning, we discard prompts that fail to demonstrate any improvement over multiple iterations to ensure effective coverage of the viable solution space; iii) leveraging **historical memory**, we provide generators with structured summaries of previous rounds and failure analyses to facilitate iterative refinement (Madaan et al., 2023; Shinn et al., 2023) and prevent error recurrence; iv) through **cross-evaluation**, we test identical prompts across heterogeneous model configurations to expose model-specific sensitivities, which provides the reward model with critical signals to distinguish the performance variations among various models for the same prompt. This iterative accumulation process forms a comprehensive offline dataset for reliable reward modeling.

### 3.3. Uncertainty-Aware Reward Model

We introduce an uncertainty-aware reward model that provides efficient and robust performance estimates for generated prompts. Taking the heterogeneous tuple $(t, m, p)$ as input, the reward model $R_\theta$ predicts a joint distribution over the set of performance objectives $\mathcal{K}$ to approximate the stochastic evaluation process:

$$R_\theta(t, m, p) = \left( (\hat{\mu}_k(t, m, p), \hat{\sigma}_k(t, m, p)) \right)_{k \in \mathcal{K}}, \quad (3)$$

where $\hat{\mu}_k(t, m, p)$ and $\hat{\sigma}_k(t, m, p)$ denote the predicted mean value that quantifies the performance score and the aleatoric uncertainty for the $k$-th objective, respectively.

The training objective prioritizes the relative ranking of predicted performance and robust evaluation over absolute numerical precision. To systematically capture the data-inherent noise (aleatoric uncertainty) across diverse task and model configurations, we leverage a Gaussian Negative Log-Likelihood (NLL) loss (Kendall & Gal, 2017; Lakshminarayanan et al., 2017). This term serves as an uncertainty-aware regularizer that mitigates the impact of outliers or high-entropy samples arising from the intrinsic stochasticity of LLM text generation. Specifically, by assigning larger aleatoric variances to samples with highly fluctuating evaluation scores, the model effectively attenuates the penalty associated with these noisy instances, thereby focusing optimization on learning reliable ranking patterns. For each objective $k$ and sample $i$, the Gaussian NLL is defined as:

$$\mathcal{L}_{\text{NLL},k}^{(i)} = \frac{(y_k^{(i)} - \hat{\mu}_k^{(i)})^2}{2\hat{\sigma}_k^{(i)2}} + \frac{1}{2} \log \hat{\sigma}_k^{(i)2} + \frac{1}{2} \log(2\pi), \quad (4)$$

where the term $\frac{1}{2} \log(2\pi)$ ensures that the likelihood is properly normalized. We further stabilize the mean-value estimate by incorporating a Mean Squared Error (MSE) term, ensuring that the predicted rankings align with the ground-truth performance distribution. We calculate the unified loss

function $\mathcal{L}_k$ for each objective $k$ as follows:

$$\mathcal{L}_k = \frac{1}{N} \sum_{i=1}^{N} \left[ \lambda_{\text{MSE}} (y_k^{(i)} - \hat{\mu}_k^{(i)})^2 + \lambda_{\text{NLL}} \mathcal{L}_{\text{NLL},k}^{(i)} \right], \quad (5)$$

where $N$ is the number of training samples, and $\lambda_{\text{MSE}}$ and $\lambda_{\text{NLL}}$ serve as hyperparameters that modulate the focus between mean estimation and variance alignment. The overall objective aggregates these losses across all objectives as:

$$\mathcal{L}_{\text{reward}} = 1/|\mathcal{K}| \cdot \sum_{k \in \mathcal{K}} \mathcal{L}_k. \quad (6)$$

### 3.4. Multi-Objective Conditional Policy

We propose a two-stage framework to train the multi-objective conditional policy. The objective is to generate prompts that lie on the Pareto frontier of quality, latency, and token consumption. The framework consists of SFT with curriculum learning (Bengio et al., 2009) for initialization and multi-objective RL using GRPO.

**Supervised Fine-Tuning.** We initialize the policy network using SFT to ensure the generation of syntactically correct and contextually relevant prompts. A pre-trained language model serves as the backbone, utilizing Low-Rank Adaptation (LoRA) (Hu et al., 2022) to reduce memory overhead. To provide a robust initialization, we construct a high-quality instruction dataset, $\mathcal{D}_{sft}$, by filtering the prompts from the heterogeneity-aware dataset described in Section 3.2 to retain only those that satisfy a threshold. The model parameters are optimized via the cross-entropy loss function:

$$\mathcal{L}_{CE}(\phi) = -\mathbb{E}_{\mathcal{D}_{sft}} \left[ \sum_{l=1}^{L} \log \pi_\phi(p_l | p_{<l}, t, m) \right], \quad (7)$$

where $L$ is the length of the prompt sequence and $p_l$ denotes the $l$-th token. We incorporate curriculum learning to train the model on simple logical structures before gradually exposing the policy to complex tasks. This strategy can effectively prevent the model from collapsing into suboptimal solutions during the early stages of optimization.

**Multi-Objective Reinforcement Learning.** We advance policy training using GRPO to explore the Pareto frontier. The policy is conditioned on a preference vector $\mathbf{w} \sim \text{Dir}(\alpha)$ to uniformly cover the multi-objective trade-off space. For an input $(t, m, \mathbf{w})$, the policy generates candidate prompts $\{p_1, \ldots, p_G\}$ evaluated by $R_\theta$. Crucially, to prevent reward hacking on out-of-distribution prompts, we extract epistemic uncertainty via Monte Carlo Dropout (Gal & Ghahramani, 2016). By executing $M$ stochastic forward passes with active dropout, we obtain the predicted mean $\hat{\mu}_{k,j}$ for objective $k$ at pass $j$. We then define the consensus mean $\mu_k = \frac{1}{M} \sum_{m=1}^{M} \hat{\mu}_{k,j}$ and the decoupled epistemic uncertainty $\sigma_k = \sqrt{\frac{1}{M} \sum_{j=1}^{M} (\hat{\mu}_{k,j} - \mu_k)^2}$. The final prompt

reward $R$ incorporates this explicit epistemic penalty to mitigate overestimation:

$$R(t, m, p, \mathbf{w}) = \sum_{k \in \mathcal{K}} w_k (\mu_k - \lambda \sigma_k), \quad (8)$$

where $\lambda$ controls the degree of conservatism, and $(\mu_k, \sigma_k)$ is standardized via z-score normalization based on their raw MC Dropout estimates. The advantage $A_i$ for the $i$-th prompt is computed by standardizing the rewards within each group, which effectively reflects the relative quality of each prompt compared to its peers. The policy is updated by maximizing the GRPO objective:

$$r_i = \pi_\phi(p_i | t, m, \mathbf{w}) / \pi_{\text{old}}(p_i | t, m, \mathbf{w}), \quad (9)$$

$$J_i = \min (r_i A_i, \text{clip}(r_i, 1 \pm \epsilon) A_i), \quad (10)$$

$$\mathcal{J}_G = \mathbb{E} \left[ \frac{1}{G} \sum_{i=1}^{G} (J_i - \beta_{kl} D_{KL}(\pi_\phi || \pi_{ref})) \right], \quad (11)$$

where $r_i$ is the probability ratio between the current policy $\pi_\phi$ and sampling policy $\pi_{old}$. The term $J_i$ denotes the clipped surrogate objective, which stabilizes training by limiting the policy update step size via the clipping hyperparameter $\epsilon$. The final objective $\mathcal{J}_G$ balances reward maximization against the deviation from the reference model $\pi_{ref}$, measured by the Kullback-Leibler (KL) divergence.

### 3.5. Theoretical Analysis

**Conservative Reward Under Uncertainty.** We introduce an uncertainty penalty in Eq. (8) to establish a pessimistic lower bound on performance, effectively safeguarding the policy against reward hacking during the RL phase. To theoretically justify this mechanism, we adopt a risk-sensitive optimization framework inspired by offline RL (Yu et al., 2020; Kidambi et al., 2020). For notational simplicity, we suppress the conditioning on the task $t$, model $m$, and preference vector $\mathbf{w}$, focusing instead on the scalarized reward induced by a fixed $(t, m, \mathbf{w})$. We assume that the true reward $R^*(p)$ is bounded by $R_{\max}$ and that the prediction error follows a sub-Gaussian distribution. Specifically, we let $\beta$ be a scaling constant such that the estimation error satisfies $|R^*(p) - \mu(p)| \leq \beta \sigma(p)$ with a failure probability of at most $\delta$, as detailed in Appendix A.

**Theorem 3.1.** *Let* $J(\pi) = \mathbb{E}_{p \sim \pi}[R^*(p)]$ *be the true expected performance and* $J_{pen}(\pi) = \mathbb{E}_{p \sim \pi}[\mu(p) - \lambda \sigma(p)]$ *be the penalized objective. If the conservatism parameter satisfies* $\lambda \geq \beta$*, we have:*

$$J(\pi) \geq J_{pen}(\pi) - 2\delta R_{\max}. \quad (12)$$

Please refer to Appendix A for the proof of the theorem. According to Theorem 3.1, the error term $2\delta R_{\max}$ represents the worst-case deviation arising from failure

events of both the true reward $R^*$ and the penalized proxy $R_{pen}(p) = \mu(p) - \lambda\sigma(p)$. This result guides hyperparameter tuning, as a smaller $\delta$ value tightens the bound on the penalized objective, reducing the risk of overestimation. We satisfy the boundedness requirement through a reward normalization strategy as described in Section 3.4.

**Pareto Optimality of Scalarized Rewards.** To theoretically validate our multi-objective optimization strategy, we analyze the relationship between the scalarized reward objective and Pareto optimality. The following proposition establishes that optimizing the weighted sum of rewards guarantees a solution on the Pareto frontier.

**Proposition 3.2.** *Let* $\mathbf{w} \in \mathbb{R}^{|\mathcal{K}|}$ *be a preference vector with strictly positive components ($w_k > 0$ for all $k \in \mathcal{K}$). If a prompt $p^*$ maximizes the scalarized objective $J_{\mathbf{w}}(p) = \sum_{k\in\mathcal{K}} w_k R_k(p)$, $p^*$ is a properly Pareto-optimal solution.*

Please refer to Appendix B for the detailed proof based on Geoffrion's theorem (Geoffrion, 1968). According to Proposition 3.2, maximizing the weighted sum acts as a sufficient condition to achieve Pareto optimality. This provides the theoretical basis for our training strategy, where sampling weights $w$ from a Dirichlet distribution, TAMPO effectively explores diverse directions in the objective space.

### 3.6. Implementation of TAMPO

Algorithm 1 details the implementation of the TAMPO training pipeline. The framework operates in three sequential stages to transform offline data into a controllable, multi-objective conditional policy. In **Stage 1**, we train the uncertainty-aware reward model $R_\theta$ on the offline dataset $\mathcal{D}$. This involves minimizing hybrid loss (Eq. 6) to accurately predict both performance metrics and their associated epistemic uncertainty. **Stage 2** warms up the policy $\pi_\phi$ via SFT. To handle the complexity of conditional generation, we adopt a curriculum learning strategy that progressively updates the model on data subsets of increasing difficulty ($\mathcal{D}_{sft}^{(c)}$), ensuring the policy masters basic instruction following before tackling complex optimization tasks. **Stage 3** performs the final optimization via multi-objective RL. In each iteration, we sample a batch of tasks and dynamic preference vectors $\mathbf{w} \sim \text{Dir}(\alpha)$ to generate candidate prompt groups. The raw predictions from the frozen reward model are normalized to a unified scale and combined into a conservatism-aware reward. Finally, the policy parameters are updated using the GRPO objective to maximize this reward while maintaining training stability through KL regularization with respect to the reference model.

## 4. Experiments

To evaluate the effectiveness and efficiency of TAMPO, we implemented the framework on top of PyTorch. All

---

**Algorithm 1** TAMPO Training Pipeline

**Input:** i) an offline dataset $\mathcal{D}$, a SFT dataset $\mathcal{D}_{sft}$; ii) $R_\theta, \pi_\phi$, initialized models; iii) $N_b, G$, batch & group sizes; iv) $\alpha, \beta_{kl}, \lambda, \iota$, hyperparameters; v) $C$, curriculum levels;
**Output:** $\pi_\phi^*$, optimized policy;
1: */* Stage 1: Reward Modeling (Section 3.3) */*
2: **while** not converged **do**
3: $\quad \mathcal{B} \leftarrow \text{Sample}(\mathcal{D}, N_b)$
4: $\quad \{\hat{\mu}^{(i)}, \hat{\sigma}^{(i)}\}_{i=1}^{N_b} \leftarrow R_\theta(\mathcal{B})$
5: $\quad \mathcal{L}_{\text{reward}} \leftarrow \mathcal{L}_{MSE}(\hat{\mu}, y) + \mathcal{L}_{NLL}(\hat{\mu}, \hat{\sigma}, y)$ *// Eq. 6*
6: $\quad \theta \leftarrow \theta - \eta_1 \nabla_\theta \mathcal{L}_{\text{reward}}$
7: **end while**
8: */* Stage 2: SFT Initialization (Section 3.4) */*
9: **for** $c = 1$ to $C$ **do**
10: $\quad \mathcal{D}_c \leftarrow \text{GetSubset}(\mathcal{D}_{sft}, c)$
11: $\quad$ **while** not converged **do**
12: $\quad\quad \mathcal{B}_c \leftarrow \text{Sample}(\mathcal{D}_c, N_b)$
13: $\quad\quad \phi \leftarrow \phi - \eta_2 \nabla_\phi \mathcal{L}_{CE}(\mathcal{B}_c)$ *// Eq. 7*
14: $\quad$ **end while**
15: **end for**
16: */* Stage 3: Multi-Objective RL (Section 3.4) */*
17: $\pi_{ref} \leftarrow \pi_\phi$
18: **while** not converged **do**
19: $\quad \pi_{old} \leftarrow \pi_\phi$
20: $\quad \mathcal{B} \leftarrow \text{Sample}(\mathcal{D}, N_b)$
21: $\quad \mathcal{J}_{total} \leftarrow 0$
22: $\quad$ **for** each $(t, m) \in \mathcal{B}$ **do**
23: $\quad\quad \mathbf{w} \sim \text{Dir}(\alpha)$
24: $\quad\quad \{p_1, \ldots, p_G\} \sim \pi_{old}(\cdot | t, m, \mathbf{w})$
25: $\quad\quad$ **for** $i = 1$ to $G$ **do**
26: $\quad\quad\quad \mu_k \leftarrow \text{Mean}(\{\hat{\mu}_{k,j}\}_{j=1}^M), \sigma_k \leftarrow \text{Std}(\{\hat{\mu}_{k,j}\}_{j=1}^M)$
27: $\quad\quad\quad \mu_k, \sigma_k \leftarrow \text{Normalize}(\mu_k, \sigma_k)$
28: $\quad\quad\quad R_i \leftarrow \sum_k w_k(\mu_k - \lambda\sigma_k)$ *// Eq. 8*
29: $\quad\quad$ **end for**
30: $\quad\quad A_i \leftarrow (R_i - \text{mean}(R_{1:G}))/(\text{std}(R_{1:G}) + \iota)$
31: $\quad\quad \mathcal{J}_{total} \leftarrow \mathcal{J}_{total} + \mathcal{J}_G(\phi, \pi_{ref}, A)$ *// Eq. 11*
32: $\quad$ **end for**
33: $\quad \phi \leftarrow \phi + \eta_3 \nabla_\phi(N_b^{-1} \mathcal{J}_{total})$
34: **end while**

---

experiments were conducted on a workstation equipped with dual NVIDIA RTX 4090 GPUs. Please refer to Appendix D for more experimental implementations.

### 4.1. Experimental Setup

**Evaluation Benchmark.** We established a diverse evaluation benchmark that includes MMLU (Hendrycks et al., 2021), GSM8K (Cobbe et al., 2021), BBH (Suzgun et al., 2023), and CNN/DailyMail (Hermann et al., 2015), consisting of 86 distinct tasks. We categorized these tasks into three primary domains: i) reasoning, ii) world knowledge, and iii) natural language understanding, while the detailed taxonomy is provided in the Appendix F. To construct an offline, heterogeneity-aware dataset for training the reward model, we extracted 1% subsamples from each task set. In the evaluation phase, we strictly utilized the held-out instances from these sources. Meanwhile, to evaluate TAMPO's generalization capability, we incorporated AQuA (Ling et al., 2017) in the generalization experiment to rigorously assess its robustness on unseen tasks.

*Table 1.* Comprehensive performance comparison across three domains in the evaluation benchmark.

| Method | Reasoning | | | | | World Knowledge | | | | | Natural Language Understanding | | | | | |
|---|---|---|---|---|---|---|---|---|---|---|---|---|---|---|---|---|
| | Accuracy (%) | | | Overhead | | Accuracy (%) | | | Overhead | | ROUGE-L | | BERTScore F1 | | Overhead | |
| | Avg. | Worst | $\Delta_b$ | Latency | Tokens | Avg. | Worst | $\Delta_b$ | Latency | Tokens | Avg. | $\Delta_b$ | Avg. | $\Delta_b$ | Latency | Tokens |
| Static | 40.0 | 28.4 | 8.4 | 1688 | 131 | 49.6 | 34.8 | 8.2 | 1265 | 100 | 25.3 | 2.5 | 27.0 | 4.0 | 948 | 83 |
| Static* | 36.8 | 23.5 | 11.6 | 1509 | 101 | 46.1 | 25.7 | 11.7 | 1311 | 81 | - | - | - | - | - | - |
| APE | 47.4 | 36.9 | 1.0 | 2098 | 85 | 55.7 | 41.1 | 2.1 | 2052 | 85 | 26.3 | 1.5 | 28.0 | 3.0 | 1818 | 80 |
| OPRO | 46.9 | 36.7 | 1.5 | 9236 | 92 | 53.6 | 41.1 | 4.2 | 8485 | 89 | 26.8 | 1.0 | 28.4 | 2.6 | 8766 | 79 |
| RLPrompt | 46.6 | 37.4 | 1.8 | 1400 | 110 | 52.1 | 40.2 | 5.7 | 1326 | 97 | 27.6 | 0.2 | 29.8 | 1.2 | 863 | 77 |
| TAMPO[1] | 47.8 | **39.8** | 0.6 | 304 | **21** | 57.4 | 40.6 | 0.4 | 285 | 19 | **27.8** | **0.0** | 30.2 | 0.8 | **835** | **74** |
| TAMPO[2] | 48.0 | 39.0 | 0.4 | **285** | 23 | 56.9 | 40.2 | 0.9 | **268** | **17** | 27.1 | 0.7 | 30.1 | 0.9 | 837 | 75 |
| TAMPO[3] | **48.4** | 39.7 | **0.0** | 858 | 83 | **57.8** | **41.3** | **0.0** | 396 | 28 | **27.8** | **0.0** | **31.0** | **0.0** | 855 | 76 |

The symbol "*" denotes the few-shot setting and "$\Delta_b$" represents the gap to the best average value.

**Heterogeneous Model Settings.** To emulate model-heterogeneity auto-routing environments, we selected Qwen (0.5B, 7B) and Llama (1B, 8B) as representative backbones to cover diverse architecture families and parameter scales. Meanwhile, to evaluate TAMPO's generalization capability, we incorporated Gemma-1B in the generalization experiment. We deploy each model in two quantization variants, namely FP16 and INT8, to reflect resource-constrained scenarios. This combination yields 8 distinct execution configurations, creating a challenging landscape to verify the adaptivity of APO methods.

**Baseline Settings.** We compared TAMPO against three distinct categories of APO approaches to systematically validate its effectiveness. First, we employed **Static Strategies**, including Zero-shot and Few-shot prompting, in which fixed templates serve as a lower bound on performance. Second, we included **Iteration-based Methods**, specifically APE (Zhou et al., 2023) and OPRO (Yang et al., 2024), which iteratively refine prompts through model calls but incur significant computational overhead. Third, we implemented **Training-based Methods**, selecting RL-Prompt (Deng et al., 2022) as a representative baseline that trains a policy network to generate prompts.

**Evaluation Metrics.** We employed a multi-dimensional protocol to assess the trade-off between effectiveness and efficiency. We report response quality to validate the model's output against the ground truth. Specifically, we used Accuracy for reasoning and world knowledge tasks. For natural language understanding tasks, we employed ROUGE-L (Lin, 2004) to capture lexical overlap, and BERTScore F1 (Zhang et al., 2020) to evaluate semantic similarity. Furthermore, we quantified end-to-end Latency, which consistently captures the wall-clock time for both prompt generation and target model evaluation.

### 4.2. Performance Comparison

**Analysis of Robustness across Tasks.** We focused on evaluating the robustness of prompt behaviors across varying execution contexts, as assessing prompt quality based on a fixed model configuration is inadequate under auto-routing

inference. Table 1 presents comprehensive results across the reasoning, world knowledge, and Natural Language Understanding (NLU) tasks. We categorized TAMPO into three variants based on distinct preference vectors: i) TAMPO[1] prioritizes token efficiency, ii) TAMPO[2] emphasizes low latency, and iii) TAMPO[3] focuses on response quality. In the reasoning domain, TAMPO[3] achieves the highest average accuracy of 48.4% while TAMPO[1] reduces token consumption to 21. We observe that the latency-aware variant achieves an inference time of 285 ms, whereas iterative methods such as APE incur delays of over 2000 ms. These experimental results confirm that our conditional policy enables flexible adjustments to meet specific deployment constraints. In world knowledge tasks, TAMPO[3] achieves a peak accuracy of 57.8%, a significant gain over the Static baseline. TAMPO[2] maintains a competitive accuracy of 56.9% while requiring only 268 ms of latency. For the NLU tasks, the quality-focused variant achieves a BERTScore F1 of 31.0 and a ROUGE-L score of 27.8. While training-based methods like RLPrompt achieve high accuracy in some scenarios, their worst-case reasoning performance drops to 37.4%, exposing sensitivity to model variations. We prevented such regressions by incorporating model metadata, as TAMPO[3] sustained a worst-case accuracy of 39.7%. Iterative methods like OPRO incur substantial computational overhead from repeated model invocations, whereas our approach generates prompts in a single forward pass. These results confirm that TAMPO extends beyond single-model optimization to produce routing-invariant prompts that ensure reliable performance and efficiency across dynamic execution contexts.

**Analysis of Robustness across Models.** We investigated the robustness of the generated prompts across distinct model architectures and quantization levels, as detailed in Table 2. TAMPO maintains high stability during transitions from full-precision to quantized models. Specifically, the quality-focused variant of TAMPO exhibits negligible performance degradation when quantizing from FP16 to INT8, which matches the stability of the Static baseline. In contrast, RLPrompt experiences a substantial performance loss, with a much larger average decrease in score under the same

*Table 2.* Comprehensive performance comparison across heterogeneous model architectures for both reasoning/world knowledge and natural language understanding tasks.

| Method | Reasoning and World Knowledge | | | | | | | | | | | | Natural Language Understanding | | | | | | | | | | | |
|---|---|---|---|---|---|---|---|---|---|---|---|---|---|---|---|---|---|---|---|---|---|---|---|---|
| | Llama-1B | | | Llama-8B | | | Qwen-0.5B | | | Qwen-7B | | | Llama-1B | | | Llama-8B | | | Qwen-0.5B | | | Qwen-7B | | |
| | Acc. | Lat. | Tok. | Acc. | Lat. | Tok. | Acc. | Lat. | Tok. | Acc. | Lat. | Tok. | R/B | Lat. | Tok. | R/B | Lat. | Tok. | R/B | Lat. | Tok. | R/B | Lat. | Tok. |
| *FP16 Half-Precision Performance* | | | | | | | | | | | | | | | | | | | | | | | | |
| Static | 31.6 | 894 | 135 | 51.1 | 2847 | 124 | 34.6 | 380 | 88 | 54.8 | 2825 | 114 | 25/26 | 415 | 82 | 26/29 | 1843 | 79 | 25/25 | 415 | 89 | 25/28 | 1971 | 83 |
| Static* | 24.6 | 799 | 150 | 44.5 | 2850 | 109 | 35.4 | 139 | 19 | 59.1 | 3667 | 86 | - | - | - | - | - | - | - | - | - | - | - | - |
| APE | 41.4 | 1187 | 114 | 62.7 | 4629 | 91 | 39.3 | 895 | 80 | 63.2 | 2662 | 56 | 26/27 | 804 | 80 | 27/30 | 3494 | 75 | 26/26 | 792 | 86 | 26/29 | 3784 | 80 |
| OPRO | **41.8** | 4451 | 88 | 60.9 | 18249 | 88 | 38.9 | 5165 | 92 | 59.8 | 12401 | 88 | 26/27 | 4122 | 80 | 28/30 | 16334 | 71 | 26/26 | 3864 | 84 | 27/29 | 18563 | 79 |
| RLPrompt | 40.5 | 883 | 161 | 57.3 | 3188 | 122 | 39.3 | 565 | 68 | 64.2 | 1877 | 65 | 26/29 | 404 | 81 | 27/30 | 1693 | 73 | **29/29** | 381 | 82 | 28/32 | **1732** | 75 |
| TAMPO[1] | 41.2 | 48 | **7** | 61.8 | 334 | 15 | 40.2 | **78** | 9 | 67.3 | **943** | 40 | **27/29** | **360** | **72** | 28/32 | **1582** | 69 | 27/27 | 373 | **80** | **29/33** | 1739 | 75 |
| TAMPO[2] | 41.1 | 48 | 10 | 62.2 | **255** | **13** | 39.6 | 80 | 10 | 67.1 | 1143 | 58 | **27/29** | 374 | 75 | 27/33 | 1583 | 69 | 27/28 | **372** | **80** | 27/31 | 1737 | 75 |
| TAMPO[3] | 41.3 | 355 | 47 | **62.9** | 840 | 53 | **40.5** | 367 | 51 | **67.8** | 1278 | 66 | **27/29** | 375 | 75 | **28/34** | 1611 | 70 | 28/28 | 383 | 82 | 28/33 | 1809 | 77 |
| *INT8 Quantization Performance* | | | | | | | | | | | | | | | | | | | | | | | | |
| Static | 32.3 | 486 | 134 | 52.7 | 2146 | 126 | 34.6 | 441 | 89 | 54.6 | 1792 | 114 | 25/26 | 301 | 83 | 26/29 | 1145 | 79 | 25/25 | 323 | 89 | 25/28 | 1174 | 82 |
| Static* | 24.6 | 971 | 151 | 44.7 | 1624 | 108 | 35.7 | **127** | 19 | 59.1 | 1447 | 88 | - | - | - | - | - | - | - | - | - | - | - | - |
| APE | 41.1 | 784 | 112 | 62.5 | 2730 | 91 | 39.0 | 906 | 83 | 63.0 | 2806 | 52 | 26/27 | 580 | 81 | 27/30 | 2172 | 75 | 26/26 | 617 | 86 | 26/29 | 2297 | 80 |
| OPRO | **41.4** | 3065 | 89 | 60.8 | 12455 | 87 | 38.9 | 4754 | 88 | 59.6 | 10344 | 92 | 26/27 | 2910 | 81 | 28/30 | 10142 | 72 | 26/26 | 2911 | 83 | 27/29 | 11281 | 79 |
| RLPrompt | 39.2 | 612 | 165 | 55.2 | 1621 | 112 | 38.8 | 584 | 65 | 60.3 | 1575 | 68 | 26/28 | 283 | 80 | 28/30 | 1059 | 73 | **29/28** | 293 | 81 | 28/32 | **1058** | 74 |
| TAMPO[1] | 40.8 | 34 | **5** | 61.8 | 108 | 10 | 40.5 | 163 | 28 | 67.1 | 647 | 43 | **27/29** | **263** | **73** | 28/32 | **1001** | 69 | 27/27 | 288 | 80 | **29/33** | 1074 | 75 |
| TAMPO[2] | 40.3 | **33** | 8 | 62.3 | **96** | **9** | 40.3 | 81 | **19** | 66.7 | **474** | **32** | **27/29** | 270 | 75 | 28/33 | **1001** | 69 | 27/27 | **284** | **79** | 27/30 | 1074 | 75 |
| TAMPO[3] | 41.3 | 308 | 46 | **62.8** | 553 | 50 | **40.7** | 344 | 63 | **67.5** | 972 | 63 | **27/29** | 270 | 75 | **28/34** | 1014 | 70 | 28/28 | 292 | 81 | 28/33 | 1086 | 76 |

The symbol "*" denotes the few-shot setting. **Acc.**: Accuracy (%), **Lat.**: Latency (ms), **Tok.**: Token Consumption, **R/B**: ROUGE-L/BERTScore F1.

conditions. This is because our approach is model-aware, enabling it to generate optimized prompts tailored to specific model characteristics, whereas RLPrompt is restricted to task-based prompts. Compared to iterative methods such as APE, TAMPO achieves output quality that is comparable to or even superior to that of APE. We observed that TAMPO requires significantly lower inference latency and token counts than the iteration-based baseline. We also test TAMPO across different parameter scales within the same architecture. We find that TAMPO maintains consistently low resource consumption across model sizes. Under various configurations of preference vectors, TAMPO effectively prioritizes specific objectives during prompt generation without causing significant regressions in other performance dimensions. In auto-routing scenarios, TAMPO's robustness enables users to balance response quality, speed, and conciseness to meet their specific requirements.

### 4.3. Analysis of Controllability in Pareto Frontier

Figure 2 shows the performance and overhead trade-offs for the Qwen-7B and Llama-8B models and their quantized counterparts. The frontier formed by TAMPO strictly encompasses the solution points of static, iteration-based, and training-based methods, indicating that TAMPO achieves higher accuracy while maintaining lower latency than the baseline methods. A crucial advantage of TAMPO is its ability to explicitly control the trade-off between performance and inference overhead.

Unlike baseline approaches that yield a single fixed solution, we trained a multi-objective conditional policy that can be guided by a preference vector. We adjusted this preference

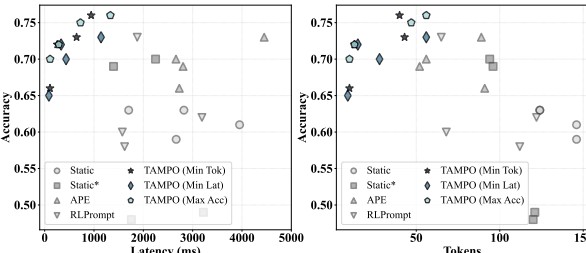

*Figure 2.* Comparison of performance and overhead trade-offs.

vector according to the task at hand: i) we assigned greater weight to accuracy when top performance is the main objective; ii) we raised the latency weight when fast inference is required; iii) we prioritized the token weight when the goal is to produce more concise outputs. Through this steering mechanism, the model can modify its behavior to meet different optimization goals. This level of controllability lets operators adjust prompt-generation preferences on the fly, without retraining, enabling dynamic adaptation to changing constraints through configurable operating modes.

### 4.4. Effectiveness of Core Components

The reliability of the proposed TAMPO framework depends on the coverage of the heterogeneity-aware dataset and the precision of the uncertainty-aware reward model. The detailed experiment settings are provided in the Appendix D.

**Dataset Coverage.** To evaluate the heterogeneity-aware dataset's coverage, Figure 3 compares TAMPO with different prompt optimization methods, including APE and OPRO. We iterated each method 10 times on the tasks "VeryHard" and "Hard" of the MMLU benchmark, gen-

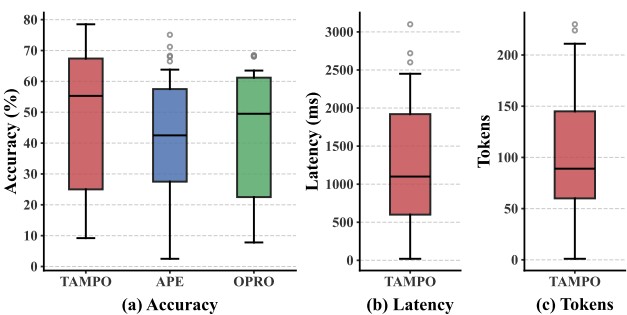

Figure 3. Comparison of coverage performance.

erating 5 prompt candidates per iteration using the GPT-5 model. As depicted in Figure 3(a), TAMPO demonstrates a higher median accuracy compared to APE and OPRO while maintaining a competitive performance envelope on the MMLU benchmark. To examine the operational efficiency, Figures 3(b) and 3(c) analyze the latency and token distributions of TAMPO, which are factors ignored by APE and OPRO. We find that TAMPO prefers generating CoT-style prompts to improve accuracy on highly difficult tasks. This comprehensive analysis confirms that our dataset provides effective coverage for evaluating prompt performance across multiple dimensions.

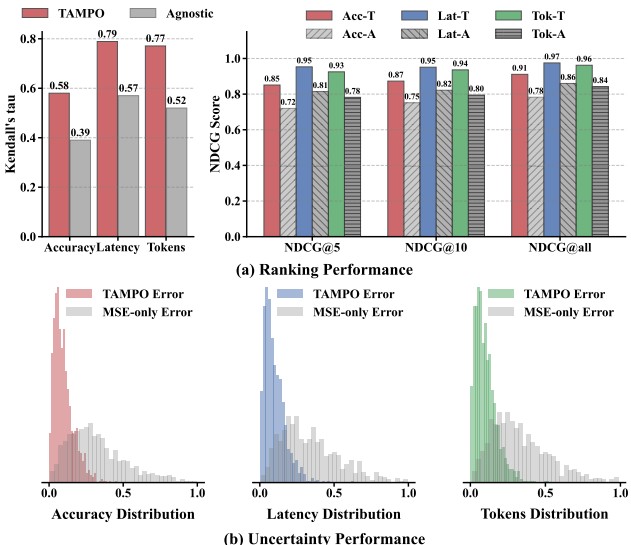

Figure 4. Comparison of performance distribution.

**Reward Model Performance.** To evaluate the reliability of the proposed uncertainty-aware reward model, we examined its ranking performance and uncertainty calibration on the heterogeneity-aware dataset. Figure 4 compares TAMPO with an Agnostic baseline without model awareness across multiple performance objectives. As depicted in Figure 4(a), TAMPO achieves significantly higher $Kendall's tau$ coefficients and NDCG scores than the model-agnostic counterpart. This performance gap confirms that incorporating model metadata is essential for accurate prompt evaluation in auto-routing scenarios where model heterogeneity signifi-

cantly influences task outcomes. Furthermore, Figure 4(b) compares the prediction errors of TAMPO with a standard MSE-only reward model to verify the risk-awareness of the uncertainty-aware reward model. This concentration indicates that the uncertainty-aware loss function improves prediction accuracy and provides calibrated confidence intervals, helping the policy recognize high-risk zones. However, the MSE-only model lacks this critical NLL regularization during training, causing the network backbone to overfit to stochastic evaluation noise and collapse its feature representations. Consequently, its MC Dropout variance fails to accurately signify model discrepancy in out-of-distribution regions, leading to severe overconfidence and reward hacking during the RL phase.

### 4.5. Analysis of Generalization

To rigorously evaluate TAMPO's generalization capabilities, we constructed four distinct evaluation quadrants using the MMLU and AQuA benchmarks, paired with the Llama-1B and Gemma-1B models. This yields four specific evaluation scenarios: i) Seen Task + Seen Model (MMLU + Llama-1B), ii) Unseen Task + Seen Model (AQuA + Llama-1B), iii) Seen Task + Unseen Model (MMLU + Gemma-1B), and iv) Unseen Task + Unseen Model (AQuA + Gemma-1B).

Table 3. Generalization performance across novel tasks and models. "Seen" refers to data present in the offline training set, while "Unseen" denotes novel tasks (AQuA) or architectures (Gemma-1B) encountered strictly during evaluation.

| Scenario | Method | Accuracy (%) | Latency (ms) | Tokens |
|---|---|---|---|---|
| Seen Task + Seen Model (MMLU + Llama-1B) | Static | 34.5 | 874 | 162 |
| | Static* | 26.6 | 813 | 152 |
| | TAMPO | **43.5** | **46** | **7** |
| Unseen Task + Seen Model (AQuA + Llama-1B) | Static | 28.3 | 844 | 154 |
| | Static* | 20.1 | 697 | 130 |
| | TAMPO | **32.8** | **380** | **52** |
| Seen Task + Unseen Model (MMLU + Gemma-1B) | Static | 42.7 | 793 | 126 |
| | Static* | 39.5 | 50 | **6** |
| | TAMPO | **45.5** | **46** | 7 |
| Unseen Task + Unseen Model (AQuA + Gemma-1B) | Static | 26.7 | 975 | 128 |
| | Static* | 25.1 | 333 | 45 |
| | TAMPO | **30.6** | **395** | 55 |

The symbol "*" denotes the few-shot setting.

As shown in Table 3, TAMPO consistently outperforms both the Zero-shot (Static) and Few-shot (Static*) baselines across all generalization scenarios. Notably, even in the extreme quadrant where both the task and the architecture are entirely unseen (AQuA + Gemma-1B), TAMPO successfully maintains its performance advantage, achieving higher accuracy than that of baselines while ensuring low inference latency. These results confirm that TAMPO's uncertainty-aware learning effectively navigates the Pareto frontier beyond its training data, making it highly reliable for continuous deployment in evolving auto-routing systems.

## 4.6. Impact of Uncertainty Modeling

To explicitly demonstrate the necessity of uncertainty modeling, we compared a standard deterministic reward model, trained solely via MSE, with TAMPO's uncertainty-aware reward model on the MMLU benchmark. As illustrated in Table 4, an MSE-only reward model is highly susceptible to overestimation, falsely predicting a high accuracy for a degraded prompt that empirically achieves only 0.12. In a standard RL pipeline, such false positives inevitably lead to severe reward hacking. Conversely, TAMPO effectively mitigates this vulnerability by recognizing the anomalous prompt, thereby flagging it with a high predictive variance. Through our conservative penalty mechanism, the final scalarized reward is heavily suppressed to 0.37. This adjustment aligns the training signal with the prompt's actual suboptimal performance, preventing the multi-objective policy from exploiting spurious reward landscapes.

*Table 4.* Comparison of reward model predictions on a degraded prompt from the MMLU benchmark.

| RM Type | Pred. Acc. | Uncertainty | Final Reward | Actual Acc. |
|---|---|---|---|---|
| MSE-only | 0.85 | N/A | 0.85 | 0.12 |
| TAMPO (Ours) | 0.82 | 0.45 | **0.37** | 0.12 |

## 5. Related Work

**Automatic Prompt Optimization.** APO reshapes the field of prompt engineering by replacing manual trial-and-error with systematic iteration- or training-based strategies. Existing iterative methods, including APE (Zhou et al., 2023) and OPRO (Yang et al., 2024), use LLMs as optimizers to progressively improve the prompts over multiple rounds. Although these approaches achieve strong performance on certain benchmarks, their reliance on repeated model calls results in higher computational overhead and increased latency. Training-based methods such as RLPrompt (Deng et al., 2022) offer an alternative by learning a policy to generate prompts in a single forward pass. However, these prior techniques typically assume a fixed deployment setting and overlook performance fluctuations caused by model heterogeneity in auto-routing pipelines.

**RL-based Prompt Optimization.** Reinforcement learning (RL) has been used to automate prompt design by refining prompts according to reward signals. Prior RL-based methods (e.g., RLPrompt (Deng et al., 2022), TEMPERA (Zhang et al., 2022), Automate-CoT (Shum et al., 2023)) generally treat prompt optimization as a single-objective task, typically focusing on maximizing accuracy. Although this works well in controlled benchmarks, such a formulation falls short for real-world auto-routing scenarios, where latency, token usage, and prediction robustness are equally important. In addition, online RL optimization (Ouyang

et al., 2022; Ramamurthy et al., 2023) incurs substantial computational and system overhead due to frequent model switching and continuous policy updates, which is at odds with the strict low-latency constraints of LLM-based auto-routing. To address this issue, offline reward models (Everitt et al., 2017; Gao et al., 2023) have been proposed to estimate performance metrics without repeated online interactions. Although recent frameworks like MORL-Prompt (Jafari et al., 2024), ParetoPrompt (Zhao et al., 2025), and Prompt-OIRL (Sun et al., 2024) have advanced multi-objective prompt optimization, they still exhibit critical limitations in auto-routing scenarios. Specifically, MORL-Prompt and ParetoPrompt primarily focus on text-centric objectives, and Prompt-OIRL lacks cross-model awareness. Meanwhile, cost-aware methods like CAPO (Zehle et al., 2025) remain constrained by the high latency of online iterative searches. Consequently, these approaches fail to balance deployment efficiency and adaptability across heterogeneous models.

TAMPO leverages an uncertainty-aware reward model to predict multi-objective metrics and a multi-objective conditional policy to explore the Pareto frontier of quality, latency, and token consumption. Users can use a preference vector to ensure that the policy generates prompts that satisfy diverse user preferences across varying task and model conditions.

## 6. Conclusion

This paper introduces TAMPO, a novel task- and model-aware APO framework for auto-routing in LLM-based systems. Specifically, we build a heterogeneity-aware dataset to train an uncertainty-aware reward model that mitigates reward hacking and guides TAMPO to learn a multi-objective conditional policy for robust prompt generation. This policy uses a preference vector to control prompts along the Pareto frontier. Comprehensive experiments on 86 tasks show that TAMPO matches the generation quality of iteration-based methods while reducing inference latency and token usage. In the future, we plan to explore low-overhead online feedback for continual policy refinement and extend the objective space to include safety and fairness.

## Acknowledgments

This work was supported by the Natural Science Foundation of China (No. 62272170). All authors are also with the MoE Engineering Research Center of SW/HW Co-Design Technology and Application, East China Normal University.

## Impact Statement

This paper presents work whose goal is to advance the field of machine learning. There are many potential societal consequences of our work, none of which we feel must be specifically highlighted here.

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

# A. Proof of Theorem 3.1

**Assumption A.1.** Let $R^*(p)$ be the true scalarized reward under a fixed preference vector $\mathbf{w}$, and $\mu(p), \sigma(p)$ be the predicted consensus mean and epistemic uncertainty. We assume the prediction error is sub-Gaussian, such that for any prompt $p$, with probability at least $1 - \delta$:

$$|R^*(p) - \mu(p)| \leq \beta\sigma(p), \tag{13}$$

where $\beta$ is a scaling constant dependent on $\delta$.

**Assumption A.2.** The reward function is bounded, i.e., $|R^*(p)| \leq R_{\max}$ and $|\mu(p)| \leq R_{\max}$ for all $p \in \mathcal{P}$. This assumption holds naturally for metrics like quality ($\in [0, 1]$) and can be enforced for latency/tokens via normalization.

Under these assumptions, we prove that the penalized objective provides a valid lower bound on the true expected performance, subject to a bounded error term.

**Proof.** Let $\mathcal{E}_p$ denote the event $\{|R^*(p) - \mu(p)| \leq \beta\sigma(p)\}$. According to Assumption A.1, this event holds with probability $\mathbb{P}(\mathcal{E}_p) \geq 1 - \delta$. Let $\mathcal{E}_p^c$ denote the failure event.

We analyze the lower bound of the true reward $R^*(p)$ relative to the penalized reward $R_{pen}(p) = \mu(p) - \lambda\sigma(p)$ by decomposing the space using the indicator function.

**Step 1: Analysis under the Good Event ($\mathcal{E}_p$).** When $\mathcal{E}_p$ holds, and given the condition $\lambda \geq \beta$, we have:

$$R^*(p) \geq \mu(p) - \beta\sigma(p) \geq \mu(p) - \lambda\sigma(p) = R_{pen}(p). \tag{14}$$

Thus, the following inequality holds:

$$R^*(p)\mathbf{1}_{\mathcal{E}_p} \geq R_{pen}(p)\mathbf{1}_{\mathcal{E}_p}. \tag{15}$$

**Step 2: Analysis under the Failure Event ($\mathcal{E}_p^c$).** Under Assumption A.2, both rewards are bounded by $R_{\max}$. The worst-case difference occurs when $R^*(p) = -R_{\max}$ and $R_{pen}(p) = R_{\max}$. Formally:

$$R^*(p) \geq -R_{\max} \quad \text{and} \quad -R_{pen}(p) \geq -R_{\max}. \tag{16}$$

Therefore, on the failure event $\mathcal{E}_p^c$:

$$R^*(p) - R_{pen}(p) \geq -2R_{\max}. \tag{17}$$

Or equivalently:

$$R^*(p)\mathbf{1}_{\mathcal{E}_p^c} \geq (R_{pen}(p) - 2R_{\max})\mathbf{1}_{\mathcal{E}_p^c}. \tag{18}$$

**Step 3: Combining and Taking Expectation.** Summing the inequalities from both events:

$$R^*(p) = R^*(p)\mathbf{1}_{\mathcal{E}_p} + R^*(p)\mathbf{1}_{\mathcal{E}_p^c} \tag{19}$$

$$\geq R_{pen}(p)\mathbf{1}_{\mathcal{E}_p} + (R_{pen}(p) - 2R_{\max})\mathbf{1}_{\mathcal{E}_p^c} \tag{20}$$

$$= R_{pen}(p)(\mathbf{1}_{\mathcal{E}_p} + \mathbf{1}_{\mathcal{E}_p^c}) - 2R_{\max}\mathbf{1}_{\mathcal{E}_p^c} \tag{21}$$

$$= R_{pen}(p) - 2R_{\max}\mathbf{1}_{\mathcal{E}_p^c}. \tag{22}$$

Finally, taking the expectation over $p \sim \pi$:

$$J(\pi) = \mathbb{E}[R^*(p)] \geq \mathbb{E}[R_{pen}(p)] - 2R_{\max}\mathbb{E}[\mathbf{1}_{\mathcal{E}_p^c}] \tag{23}$$

$$= J_{pen}(\pi) - 2R_{\max}\mathbb{P}(\mathcal{E}_p^c) \tag{24}$$

$$\geq J_{pen}(\pi) - 2\delta R_{\max}. \tag{25}$$

The last step follows from $\mathbb{P}(\mathcal{E}_p^c) \leq \delta$. This completes the proof. $\square$

# B. Proof of Pareto Optimality

In this section, we provide the theoretical justification for the design choice of using randomized linear scalarization in TAMPO. We ground our analysis in the classic multi-objective optimization theory established by Geoffrion (1968).

## B.1. Problem Formulation

Let the vector-valued reward function be $\mathbf{R}(p) = [R_1(p), \ldots, R_k(p)]^T$, where $p \in \mathcal{P}$ denotes a prompt in the discrete search space. We define the problem as finding solutions on the *Pareto Frontier*. A solution $p^*$ is *Pareto optimal* if there exists no $p' \in \mathcal{P}$ such that:

$$\forall i \in \{1, \ldots, k\}, R_i(p') \geq R_i(p^*) \quad \text{and} \quad \exists j, R_j(p') > R_j(p^*) \tag{26}$$

## B.2. Proposition 1

**Proposition 1 (The Weighted-sum optimality implies the proper Pareto optimality).** *Let* $\mathbf{w} \in \mathbb{R}^k$ *be a weight vector with strictly positive components* $(w_i > 0, \forall i)$. *If a prompt* $p^*(\mathbf{w})$ *is a global maximizer of the scalarized objective:*

$$p^*(\mathbf{w}) \in \arg\max_{p \in \mathcal{P}} J_{\mathbf{w}}(p) = \arg\max_{p \in \mathcal{P}} \sum_{i=1}^{k} w_i R_i(p) \tag{27}$$

*then* $p^*(\mathbf{w})$ *is a proper Pareto-optimal solution.*

## B.3. Detailed Proof

We prove this proposition by contradiction.

**Assumption:** Assume that $p^*$ maximizes the scalarized objective function $J_{\mathbf{w}}(p)$ with strictly positive weights $\mathbf{w} > \mathbf{0}$, but $p^*$ is **not** Pareto optimal.

**Step 1: Implication of Non-Optimality** If $p^*$ is not Pareto optimal, by definition, there must exist a dominating solution $p' \in \mathcal{P}$ such that:

$$R_i(p') \geq R_i(p^*) \quad \forall i = 1, \ldots, k \tag{28}$$
$$R_j(p') > R_j(p^*) \quad \text{for at least one index } j \tag{29}$$

**Step 2: Weighted Aggregation** Since all weights are strictly positive ($w_i > 0$), multiplying the inequalities by $w_i$ preserves their direction. For the index $j$ where the strict inequality holds (Eq. 29):

$$w_j R_j(p') > w_j R_j(p^*) \tag{30}$$

For all other indices $i \neq j$ (Eq. 28):

$$w_i R_i(p') \geq w_i R_i(p^*) \tag{31}$$

**Step 3: Summation and Contradiction** We verify the scalarized reward sum for the dominant solution $p'$:

$$J_w(p') = \sum_{i=1}^{k} w_i R_i(p') = w_j R_j(p') + \sum_{i \neq j} w_i R_i(p') \tag{32}$$

Substituting the inequalities from Step 2:

$$J_w(p') > w_j R_j(p^*) + \sum_{i \neq j} w_i R_i(p^*) = \sum_{i=1}^{k} w_i R_i(p^*) \tag{33}$$

Thus, we derive:

$$J_w(p') > J_w(p^*) \tag{34}$$

This inequality implies that $p^*$ is not the maximizer of the scalarized objective $J_w(\cdot)$, which contradicts our initial premise.

**Conclusion:** The assumption that $p^*$ is not Pareto optimal must be false. Therefore, $p^*$ is Pareto optimal. Furthermore, since $\mathbf{w} > \mathbf{0}$, $p^*$ satisfies the conditions for proper efficiency. $\square$

## C. The Details of Heterogeneity-Aware Dataset Construction

To ensure the multi-objective conditional policy possesses robust generalization capabilities and model-aware adaptability, we meticulously constructed a heterogeneity-aware prompt dataset. This section elaborates on the experimental configuration and the four core iterative generation strategies introduced in Section 3.2.

### C.1. Experimental Configuration and Sampling

All evaluation experiments were conducted on a local workstation equipped with dual NVIDIA RTX 4090 GPUs. To maintain a balanced representation across diverse benchmarks while mitigating computational overhead, we adopted a 1% subsampling strategy from each source's set to construct the evaluation foundation.

To simulate realistic auto-routing environments, our dataset spans a broad range of model architectures and parameter scales, specifically featuring the Qwen family (Qwen2.5-0.5B-Instruct and Qwen2.5-7B-Instruct) and the Llama family (Llama-3.2-1B-Instruct and Llama-3.1-8B-Instruct). These models were sourced from Hugging Face in the GGUF format and deployed locally using the `llama.cpp` framework to perform both quantization and inference. Each model was evaluated across two distinct quantization variants, including half-precision FP16 and 8-bit INT8 (q8_0), to provide the reward model with a comprehensive view of the solution space with respect to accuracy, latency, and token consumption.

### C.2. Iterative Strategies

The construction of the heterogeneity-aware prompt dataset follows an iterative refinement process governed by four core mechanisms:

**Style Diversity.** To ensure diversity in the generated prompts and prevent stylistic convergence, we used GPT-4, GPT-4o, and GPT-5 as the primary prompt generators. We explicitly incorporated style-diversity guidance into the meta-prompts, instructing the generators to explore distinct prompting paradigms, such as zero-shot, few-shot, role-playing, and Chain-of-Thought (CoT). Furthermore, to enhance efficiency, the generator produces five distinct prompt candidates per round, enabling it to detect and compare different styles in a single inference step.

**Elimination Mechanism.** We defined a fixed optimization cycle of 10 rounds and introduced a multi-objective elimination mechanism.

- **Upper Threshold (Early Termination)**: In each iteration, if a prompt's performance across quality, latency, and token consumption exceeds the predefined ideal thresholds, it is considered a successful candidate, and its optimization is terminated early to avoid redundant evaluations and save computational cost.

- **Lower Threshold (Pruning)**: A "survival-of-the-fittest" check is performed at the 5th round. If a prompt's metrics fall below the minimum acceptable thresholds, it is discarded and replaced with a fresh candidate, thereby reallocating search budget toward more promising regions of the space.

These thresholds are dynamically adjusted based on task difficulty (e.g., "VeryHard" or "Easy") and on whether CoT is employed, as CoT inherently increases token consumption.

**Historical Memory.** To facilitate "learning from failure," we provided the high-capability LLM with the prompts, corresponding model responses, objective scores, and failure analyses from the previous two rounds.

- **Cost Balancing**: To control API costs, we limited the length of model responses to the first 50 characters in historical memory.

- **Variable Control**: For a given task, the first three test questions during local inference remain constant across all iterations, enabling the generator to isolate the impact of prompt variations on model behavior.

- **Deep Reasoning**: Before generating new prompts, the generator is required to output a reasoning-based analysis of historical failures, helping it diagnose the root causes of low performance and avoid recurring errors.

**Cross-Evaluation.** Cross-evaluation of the same prompt across different models for the same task provides critical signals that help the reward model distinguish model-specific sensitivities.

- **Intra-Family**: Within the same model family (e.g., Llama), full cross-evaluations are conducted across all parameter scales and quantization types.

- **Inter-Family**: To reduce time costs, cross-evaluations between different model families are restricted to FP16 variants only.

By focusing these cross-evaluations on the iteration-best prompts, we significantly reduced computational overhead while preserving the dataset's ability to capture model heterogeneity. In total, the base iterative process yielded 86 tasks $\times$ 8 models $\times$ 10 rounds $\times$ 5 candidates per round $= 34{,}400$ prompt candidates. To ensure robust performance estimates and to thoroughly capture diversity in generation, we conducted additional repeated sampling on selected configurations. After comprehensive cross-evaluation and incorporating these supplementary instances, we obtained a final heterogeneity-aware dataset of 328,400 records, each paired with its corresponding performance metrics.

### C.3. SFT Dataset $\mathcal{D}_{sft}$ Construction

As described in Section 3.4, the SFT stage relies on a high-quality instruction dataset. We construct $\mathcal{D}_{sft}$ by filtering the heterogeneity-aware dataset using a minimum reward threshold $\tau_{\min}$, retaining only samples whose scalarized reward satisfies $R \geq \tau_{\min}$.

The threshold $\tau_{\min}$ is determined through a data-driven approach. We first compute the scalarized reward $R(t, m, p, \mathbf{w})$ for all samples in the heterogeneity-aware dataset using the reward function defined in Equation (8), where we use task-specific default weights $\mathbf{w}$ (e.g., $w_{acc} = 0.6$, $w_{lat} = 0.2$, $w_{tok} = 0.2$ for open tasks). We then analyze the reward distribution and select $\tau_{\min} = 0.4$ as this threshold. This choice balances two competing objectives: i) maintaining high-quality training data that provides strong learning signals, and ii) ensuring sufficient data volume to prevent overfitting. Through empirical validation, we found that $\tau_{\min} = 0.4$ effectively filters out low-quality prompts (e.g., those with low accuracy or excessive latency) while retaining approximately 60% of the original dataset, providing adequate diversity for policy initialization.

### C.4. Prompt Templates

```
1  """task: [TASK] task_type: [TASK_TYPE]
2  model: [MODEL_NAME] ([MODEL_TYPE], [MODEL_SIZE], precision: [QUANTIZATION_PRECISION])
3
4  [full_DEMO]
5
6  Based on the above examples, task requirements, and model characteristics, generate a
       prompt that would work well for this specific task and model combination. Consider:
7  1. Task and model-specific requirements and constraints
8  2. Model capabilities and limitations
9  3. Balance among effectiveness, computational efficiency, and prompt length
10 4. Few-shot, chain-of-thought, etc. style
11 5. Clear and concise instructions without specific questions
12
13 You MUST respond with a single, valid JSON object in the following format:
14 {"is_cot": false, "generated_prompt": "[The generated prompt text goes here]"}. Set "
       is_cot" to true only if the prompt is a Chain-of-Thought style prompt."""
```

*Listing 1.* Prompt Generation Template

```
1  "Question: [INPUT]\nAnswer: [OUTPUT]"
```

*Listing 2.* Demo Template

```
1  "Instructions: [PROMPT].\n\nQuestion: [INPUT]\nAnswer: [OUTPUT]"
```

*Listing 3.* Evaluation Template

```
1  """task: [TASK] task_type: [TASK_TYPE]
2  model: [MODEL_NAME] ([MODEL_TYPE], [MODEL_SIZE], precision: [QUANTIZATION_PRECISION])
3
4  [CONTEXT]
5
6  Analyze why these prompts result in poor performance in terms of accuracy, latency, and
       token count.
7  Generate [NUM_PROMPTS] optimized prompts based on the provided information, ensuring
       concise, varied outputs (e.g., zero-shot, few-shot, chain-of-thought, and role-
       playing) with high accuracy for the specified task and model.
8  few-shot style:
9  [full_DEMO]
10
11 You MUST respond with a single, valid JSON object in the following format:
12 {
13   "analysis": "[Your detailed analysis of the poor performance goes here]",
14   "optimized_prompts": [
15     [OPTIMIZED_PROMPTS_LIST]
16   ]
17 }
18 For each prompt in "optimized_prompts", set "is_cot" to true if it is a Chain-of-
       Thought style prompt, otherwise false."""
```

*Listing 4.* Prompt Optimization Template

```
1  "[TASK_TYPE] [TASK_DESCRIPTION] [TASK_DIFFICULTY] [MODEL_NAME] [MODEL_TYPE] [MODEL_SIZE
       ] [QUANTIZATION] [PREFERENCE_VECTOR]"
```

*Listing 5.* TAMPO Template

In our framework, we initially use the **Prompt Generation Template** to produce a base set of candidate prompts. Throughout the iterative refinement process, the **Prompt Optimization Template** is used to improve these prompts by integrating performance feedback and failure analyses. To facilitate a few-shot prompting, the **Demo Template** structures in-context examples, while the **Evaluation Template** ensures a standardized format for assessing response quality and inference efficiency during evaluation. During inference, we use the **TAMPO Template** to construct the policy model's input, incorporating a preference vector to guide prompt generation toward specific objective trade-offs.

## D. Implementation Details

### D.1. Hyperparameter Settings

This section summarizes the hyperparameters used to train the uncertainty-aware reward model and the multi-objective conditional policy. Unless otherwise specified, we use a fixed random seed of $42$ throughout all experiments.

**Reward model.** We train a multi-target regression reward model based on DeBERTa-v3-Large, which predicts three performance dimensions (*accuracy*, *latency*, and *token count*). When aleatoric uncertainty prediction is enabled, the model outputs a Gaussian mean and aleatoric variance for each target and is optimized with a weighted combination of mean squared error (MSE) and Gaussian negative log-likelihood (NLL):

$$\mathcal{L}_{reward} = \lambda_{\text{MSE}} \cdot \mathcal{L}_{\text{MSE}} + \lambda_{\text{NLL}} \cdot \mathcal{L}_{\text{NLL}}, \tag{35}$$

where we set $\lambda_{\text{MSE}} = 0.5$ and $\lambda_{\text{NLL}} = 0.5$. To stabilize aleatoric uncertainty learning, we enforce a minimum variance floor $\sigma_{\min}^2$ via $\sigma^2 = \exp(\log \sigma^2) + \sigma_{\min}^2$, with $\sigma_{\min}^2 = 10^{-5}$. Crucially, during policy training and inference, we perform $M = 5$ stochastic Monte Carlo forward passes to estimate epistemic uncertainty (as detailed in Table 5). We normalize regression targets using training-set statistics (z-score) and de-normalize predictions at inference time.

**Policy Model.** Our policy is a decoder-only LM conditioned on task and model metadata (Section 3.4). We train it

*Table 5.* Reward model hyperparameters.

| Component | Setting |
|---|---|
| Backbone | DeBERTa-v3-Large |
| Max input length | 1024 |
| Targets | quality, latency, token_count |
| Target normalization | z-score on training split |
| Uncertainty prediction | enabled |
| Variance floor | $\sigma_{\min}^2 = 10^{-5}$ |
| Dropout | 0.2 |
| MC dropout passes ($M$) | 5 |
| Regression head hidden size | 512 |
| Loss weights | $\lambda_{\mathrm{MSE}} = 0.5, \lambda_{\mathrm{NLL}} = 0.5$ |
| Optimizer | AdamW |
| Learning rate | $2 \times 10^{-5}$ |
| Weight decay | 0.01 |
| LR schedule | cosine; warmup ratio 0.06 |
| Batch size (per device) | train 2 / eval 2 |
| Gradient accumulation | 4 |
| Epochs | 60 |
| Mixed precision | FP16 |

with a two-stage pipeline: i) SFT initialization with curriculum learning, ii) multi-objective RL training with GRPO. The hyperparameters of the policy are summarized in Table 6.

**Pseudo-code Details.** In our pseudo-code, we use $G$ to denote the number of candidate prompts per condition, $\epsilon$ for the GRPO clipping parameter, $\beta_{kl}$ for the KL coefficient, and $\lambda$ for the uncertainty penalty in Eq. (8). For label normalization in the reward model and reward normalization in GRPO, we use z-score normalization by default, and we summarize the concrete hyperparameter values in Table 6.

## D.2. The Details of Experiment Settings

### D.2.1. Answer Extraction and Evaluation Protocol

In all experiments, we adopt a deliberately simple answer-extraction rule for parsing model outputs, avoiding task-specific post-processing heuristics that require extensive manual tuning. While this choice improves reproducibility and lowers the engineering barrier, it may underestimate the absolute performance of some methods (e.g., those that benefit from carefully designed output-format constraints).

### D.2.2. Baseline Settings

**Static.** We use a fixed task description (instruction) and directly concatenate it with the input question for each model. Since no few-shot demonstrations are provided, the generated outputs often exhibit inconsistent formatting and may include redundant content.

**Static\*.** We augment the fixed task description with two few-shot demonstrations. This helps the model better infer the intended output format and typically yields more concise responses. However, weaker models may confuse demonstrations with the test query, occasionally copying example-related content into the final answer. Due to the additional context length and high inference cost introduced by the demonstrations, this baseline is not included in the natural language understanding comparison in Section 4.2.

**APE.** APE is an iterative prompt optimization baseline. Following our experimental budget, we run two optimization rounds; in each round, GPT-4o generates five prompt candidates, which are evaluated on all target models. We report i) the total inference latency accumulated across the two rounds and ii) the final selected prompt's quality and token cost.

**OPRO.** OPRO is also an iterative optimization method. We run 10 optimization rounds; in each round, GPT-4o generates 5 candidates, and we evaluate them across all target models. We report the cumulative inference latency over 10 rounds, as well as the quality and token cost of the final selected prompt.

*Table 6.* Policy training hyperparameters.

| Component | Setting |
|---|---|
| Backbone | Phi-3.5-mini-instruct |
| SFT dataset filter | min reward $\tau_{\min} = 0.4$ |
| LoRA | enabled; $r = 16$, $\alpha = 32$, dropout 0.1 |
| Curriculum learning | enabled; ratios $0.2\rightarrow1.0$; 7 stages; linear schedule; warmup 1 epoch |
| Diversity regularization | Self-BLEU; weight 0.2; min length 5 |
| **SFT** epochs | 6 |
| **SFT** learning rate | $3 \times 10^{-5}$ |
| **SFT** batch size | 8 (per device) |
| **SFT** warmup steps | 100 |
| **GRPO** epochs | 10 |
| **GRPO** batch size | 8 (per device) |
| **GRPO** learning rate | $1 \times 10^{-5}$ |
| Clip range | $\epsilon = 0.2$ |
| KL penalty | $\beta_{kl} = 0.05$ |
| Candidate generation | max_new_tokens 128; $T = 0.8$; top-$p$=0.9; top-$k$=50; $G = 5$ |
| Reward normalization | z-score |
| Uncertainty penalty | enabled; $\lambda = 1.0$ |
| Random scalarization | enabled; Dirichlet $\boldsymbol{\alpha} = [2, 1, 1]$ |
| Gradient accumulation | 2 |
| Max grad norm | 1.0 |
| Precision | BF16 |
| Dataloader workers | 8 |

**RLPrompt.** RLPrompt is a training-based baseline. Since it is not explicitly model-aware, we trained it using reward feedback obtained by executing prompts on a single reference model (Qwen2.5-7B-Instruct) and then evaluated the learned prompts on the remaining models to assess cross-model generalization.

**TAMPO[1].** We set the preference vector to prioritize token efficiency (i.e., emphasizing the token objective) and evaluate the resulting prompts across tasks and models.

**TAMPO[2].** We set the preference vector to prioritize low latency (i.e., emphasizing the latency objective) and evaluate across tasks and models.

**TAMPO[3].** We set the preference vector to prioritize quality/accuracy (i.e., emphasizing the quality objective) and evaluate across tasks and models.

### D.2.3. END-TO-END LATENCY PROTOCOL

We consistently calculated latency for all baselines and TAMPO. Specifically, end-to-end latency comprises two stages across all methods: prompt generation time and target model evaluation time. Because different approaches acquire optimized prompts through fundamentally different mechanisms, their generation times naturally capture different operational overheads. For training-based methods like TAMPO and RLPrompt, prompt generation time strictly refers to the time for a single forward pass of the policy network, excluding one-time model-loading overhead. For iteration-based methods like APE and OPRO, discovering the optimized prompt at runtime inherently requires accumulating the API call time across all optimization iterations. Once the prompt is acquired, the total latency for each method explicitly includes the local model inference time required to evaluate the final prompt.

### D.2.4. BENCHMARK SETTINGS

Section 4.1 details the benchmarks used in our experiments. For CNN/DailyMail, we restricted evaluation to 1,796 articles (each under 2,000 characters) to accommodate context-window and computational constraints. These examples are used for the performance comparisons reported in Section 4.2.

**Pareto-Frontier Controllability.** We conducted controllability analysis on the MMLU World Knowledge subset, which contains 57 tasks. The baselines include Static, Static*, APE, OPRO, and RLPrompt. We evaluated baselines and TAMPO on two backbone LLMs (Qwen-7B and Llama-8B), each under two inference settings (FP16 and INT8). Figure 2 omits OPRO because its 10-round iterative optimization incurred substantially higher latency, making it impractical to visualize alongside other methods. To assess controllability, we reported three preference-conditioned variants: TAMPO[1] (Min-Token), TAMPO[2] (Min-Latency), and TAMPO[3] (Max-Accuracy).

**Dataset Coverage.** To measure dataset coverage under challenging conditions, we ran each method 10 times independently on the "VeryHard" and "Hard" tasks in MMLU. In each trial, we performed 10 optimization iterations and generated 5 prompt candidates per iteration using GPT-5.

**Reward Model Performance.** For Figure 4(a), we evaluated ranking quality on the test split of the heterogeneity-aware dataset, comparing the Agnostic baseline against TAMPO. For Figure 4(b), we studied the effect of uncertainty modeling by comparing reward-model prediction errors when training with and without the NLL term (i.e., ablating $\lambda_{\mathrm{NLL}}$ in Eq. (35)).

# E. Additional Experimental Results and Analysis

## E.1. Upfront Costs and Long-Term Amortization

While TAMPO requires an upfront investment for dataset construction and policy training, this fixed cost is heavily amortized during deployment. Table 7 presents a detailed breakdown of the computational and financial costs associated with our framework.

*Table 7.* Detailed cost analysis of the TAMPO framework.

| Phase | Task / Details | Hardware / API | Time / Financial Cost |
|---|---|---|---|
| Data Generation | API calls | GPT-4/4o/5 | ∼$660 USD |
| Data Generation | Local cross-evaluation | 2× RTX 4090 (24G) | ∼31.6 hours |
| Reward Model | RM Training | 2× RTX 4090 (24G) | ∼15 hours |
| Policy Model | SFT Stage | 2× RTX 4090 (24G) | ∼12 hours |
| Policy Model | RL Stage | 2× RTX 4090 (24G) | ∼30 hours |

To assess the cost-effectiveness of this upfront investment, we compare it against iterative APO methods, whose marginal costs scale linearly with the number of tasks. For instance, assuming an iterative baseline uses the exact same configuration as our data generation (10 optimization rounds, 5 prompt candidates per round across 8 model settings), optimizing a single novel task requires approximately 400 API calls, costing roughly $7.67 USD ($660 USD / 86 tasks). In a dynamic auto-routing scenario with a continuous influx of new tasks or models, iterative methods incur repeated, expensive API fees and introduce unacceptable wall-clock latency.

In contrast, TAMPO heavily amortizes this initial cost. As demonstrated by the generalization experiment in Section 4.5, TAMPO exhibits strong zero-shot generalization. Once trained, the policy generates optimized prompts for a new task or model in a single forward pass. This incurs $0 in additional API costs and mere milliseconds of latency. Consequently, TAMPO's fixed upfront investment becomes highly cost-effective and operationally efficient over the long term.

## E.2. Qualitative Prompt Analysis

A core contribution of TAMPO is its ability to generate prompts tailored to the characteristics, capacities, and quantization levels of different target models. To validate this model-awareness, we conduct both qualitative analyses. Table 8 presents example prompts generated by TAMPO for the MMLU benchmark across four different model configurations. The results clearly demonstrate that TAMPO dynamically adapts its generation strategy to accommodate each model's unique limitations. For instance, smaller models like Qwen-0.5B (f16) receive longer templates with repeated soft constraints to prevent output drift. Furthermore, when the same model is quantized to 8-bit (q8_0), TAMPO adjusts the phrasing to emphasize analytical, evidence-based steps, thereby stabilizing the behavior of the quantized weights. In contrast, for models prone to format overfitting (e.g., Llama-1B), TAMPO enforces hard format contracts with negative constraints.

*Table 8.* Qualitative examples of TAMPO-generated prompts for the MMLU benchmark. TAMPO adaptively adjusts prompt length, constraints, and framing based on the target model's specific limitations and quantization level.

| Model Configuration | Model Limitations | Prompt Example | Prompt Characteristics |
|---|---|---|---|
| Qwen2.5-0.5B-Instruct (f16) | Needs explicit scaffolding; weak on vague instructions; lower accuracy ceiling. | "You are tasked with answering... Carefully read... Ensure your response is concise and directly selects the most accurate option." | Long template; repeats soft constraints (concise/accurate) to prevent drift. |
| Qwen2.5-0.5B-Instruct (q8_0) | Quantization shifts optimal prompts; needs clear framing. | "You are tasked with answering... Carefully analyze... select the most accurate answer based on verified global statistics..." | Stresses analysis and evidence-like phrasing to stabilize int8 generation. |
| Qwen2.5-7B-Instruct (f16) | Prone to overfitting surface patterns if format changes. | "Answer the following multiple-choice question... Provide only the letter corresponding to the correct answer." | Very short; uses minimal instructions to avoid overfitting. |
| Llama-3.2-1B-Instruct (f16) | Brittle to format drift; prone to echoing options without strict constraints. | "Strict output format... Return exactly one of these four strings... Any other output... will be graded 0." | Hard format contract; uses negative constraints to stop spurious tokens. |

### E.3. Cross-Assignment Evaluation

To verify that the model-specific prompts generated by TAMPO outperform universally applied prompts, we conducted a cross-assignment evaluation. Specifically, we used TAMPO to generate an optimized prompt for each of the four models, and subsequently evaluated every prompt across all four models. As shown in Table 9, the diagonal elements (bolded) consistently represent the optimal trade-off among accuracy, latency, and token consumption for each respective model. For example, a prompt optimized for Qwen-7B causes significant performance degradation and latency spikes when evaluated on Qwen-0.5B. This cross-assignment matrix conclusively demonstrates that TAMPO does not merely search for universally good text, but actively captures and exploits the idiosyncratic preferences of each target architecture.

*Table 9.* Cross-assignment evaluation on the MMLU benchmark. The rows represent the target model the prompt was optimized for, while the columns represent the model performing the evaluation. Metrics are reported as Accuracy (%) / Latency (ms) / Tokens.

| Source Model | Evaluating Model | | | |
|---|---|---|---|---|
| | Qwen-0.5B (f16) | Qwen-0.5B (q8_0) | Qwen-7B (f16) | Llama-1B (f16) |
| Qwen-0.5B (f16) | **45.4 / 80 / 10** | 42.2 / 40 / 10 | 71.4 / 1150 / 50 | 42.1 / 63 / 10 |
| Qwen-0.5B (q8_0) | 44.5 / 210 / 35 | **43.1 / 32 / 8** | 70.8 / 1300 / 65 | 42.2 / 81 / 12 |
| Qwen-7B (f16) | 40.2 / 1450 / 180 | 40.5 / 754 / 190 | **73.2 / 1143 / 56** | 40.4 / 112 / 16 |
| Llama-1B (f16) | 39.2 / 480 / 62 | 39.5 / 410 / 110 | 68.1 / 1139 / 50 | **43.5 / 46 / 7** |

### E.4. Performance Analysis Across Task Difficulty Tiers

To systematically investigate which types of tasks benefit most from our prompt optimization framework, we analyzed TAMPO's empirical performance gains stratified by task difficulty. The difficulty tiers (Easy, Medium, Hard, and Very Hard) are categorized by the inherent complexity and reasoning requirements of the corresponding benchmarks. As illustrated in Table 10, the magnitude of TAMPO's performance improvement is strongly positively correlated with task complexity. For "Easy" tasks, modern LLMs already possess robust zero-shot priors, rendering the gains from prompt optimization marginal. However, as task difficulty escalates, the necessity of precise, task-aware optimization becomes unequivocally clear. For "Very Hard" tasks, naive static prompting frequently fails to elicit meaningful responses. In these scenarios, TAMPO's ability to dynamically inject appropriate scaffolding, hard constraints, and multi-step reasoning structures yields critical performance surges of up to 25.8%. This analysis demonstrates that while TAMPO provides universal improvements, its value proposition is most profoundly realized in complex, highly constrained reasoning scenarios.

## F. Evaluation Benchmark Categorization Details

We provide the detailed taxonomy of the evaluation benchmark in Table 11. The tasks are categorized into three primary domains, including i) reasoning, ii) world knowledge, and iii) natural language understanding. This classification ensures a

*Table 10.* Analysis of TAMPO's performance gains across different task difficulty tiers.

| Difficulty Tier | Performance Gain | Underlying Mechanism / Analysis |
|---|---|---|
| **Easy** | $1.5\% - 3.2\%$ | Strong zero-shot priors are generally sufficient; structural optimization yields diminishing returns. |
| **Medium** | $5.4\% - 8.1\%$ | Refined framing and structural clarity significantly improve instruction-following consistency. |
| **Hard** | $9.6\% - 14.5\%$ | Optimization successfully unlocks and stabilizes multi-step reasoning capabilities. |
| **Very Hard** | $15.2\% - 25.8\%$ | Naive prompting frequently fails entirely; task-aware optimization and strict scaffolding are strictly necessary. |

comprehensive assessment of the model capabilities.

*Table 11.* The comprehensive taxonomy and task breakdown of the evaluation benchmark, comprising 86 unique tasks that are grouped into the domains of reasoning, world knowledge, and Natural Language Understanding (NLU).

| Cat. | Benchmark | # Tasks | Task Composition |
|---|---|---|---|
| Reasoning | BBH (Suzgun et al., 2023) | 27 | Boolean Expressions, Causal Judgement, Date Understanding, Disambiguation QA, Dyck Languages, Formal Fallacies, Geometric Shapes, Hyperbaton, Logical Deduction (Five Objects), Logical Deduction (Seven Objects), Logical Deduction (Three Objects), Movie Recommendation, Multistep Arithmetic Two, Navigate, Object Counting, Penguins in a Table, Reasoning about Colored Objects, Ruin Names, Salient Translation Error Detection, Snarks, Sports Understanding, Temporal Sequences, Tracking Shuffled Objects (Five Objects), Tracking Shuffled Objects (Seven Objects), Tracking Shuffled Objects (Three Objects), Web of Lies, Word Sorting |
| | GSM8K (Cobbe et al., 2021) | 1 | GSM8K |
| World Knowledge | MMLU (Hendrycks et al., 2021) | 57 | Abstract Algebra, Anatomy, Astronomy, Business Ethics, Clinical Knowledge, College Biology, College Chemistry, College Computer Science, College Mathematics, College Medicine, College Physics, Computer Security, Conceptual Physics, Econometrics, Electrical Engineering, Elementary Mathematics, Formal Logic, Global Facts, High School Biology, High School Chemistry, High School Computer Science, High School European History, High School Geography, High School Government and Politics, High School Macroeconomics, High School Mathematics, High School Microeconomics, High School Physics, High School Psychology, High School Statistics, High School US History, High School World History, Human Aging, Human Sexuality, International Law, Jurisprudence, Logical Fallacies, Machine Learning, Management, Marketing, Medical Genetics, Miscellaneous, Moral Disputes, Moral Scenarios, Nutrition, Philosophy, Prehistory, Professional Accounting, Professional Law, Professional Medicine, Professional Psychology, Public Relations, Security Studies, Sociology, US Foreign Policy, Virology, World Religions |
| NLU | CNN/DailyMail (Hermann et al., 2015) | 1 | CNN/DailyMail |
| **Total** | | **86** | - |

*Table 12.* Detailed descriptions and difficulty levels of tasks within the evaluation benchmarks. This table spans multiple pages and lists the source benchmark, the specific task name, the assigned difficulty level, and a functional description for each task.

| Benchmark | Task | Difficulty | Task Description |
|---|---|---|---|
| | boolean_expressions | VeryHard | Evaluate logical expressions and determine if they evaluate to True or False. |
| | causal_judgement | VeryHard | Identify the most likely cause or effect in given scenarios. |
| | date_understanding | VeryHard | Answer questions about relative dates and temporal relationships. |
| | disambiguation_qa | VeryHard | Resolve ambiguous statements by selecting the most likely interpretation. |
| | dyck_languages | VeryHard | Determine if sequences of brackets are properly nested and balanced. |
| | formal_fallacies | VeryHard | Identify whether given arguments contain formal logical fallacies. |
| | geometric_shapes | VeryHard | Answer questions about geometric shapes and their properties. |
| | hyperbaton | VeryHard | Rearrange sentences to restore normal word order from hyperbaton constructions. |
| | logical_deduction_five | VeryHard | Solve logic puzzles involving five objects and their relationships. |
| | logical_deduction_seven | VeryHard | Solve logic puzzles involving seven objects and their relationships. |
| | logical_deduction_three | VeryHard | Solve logic puzzles involving three objects and their relationships. |
| | movie_recommendation | VeryHard | Provide movie recommendations based on given preferences. |
| BBH | multistep_arithmetic_two | VeryHard | Solve arithmetic problems requiring multiple calculation steps. |

Continued on next page

**Table 12 – Continued from previous page**

| Benchmark | Task | Difficulty | Task Description |
|-----------|------|------------|------------------|
| | navigate | VeryHard | Provide navigation instructions based on spatial relationships. |
| | object_counting | VeryHard | Count objects described in text scenarios. |
| | penguins_in_a_table | VeryHard | Answer questions by extracting and processing data from tables about penguins. |
| | reasoning_about_colored | VeryHard | Answer questions about objects and their colors based on given constraints. |
| | ruin_names | VeryHard | Select the humorous edit that ruins movie or artist names through clever wordplay. |
| | translation_error | VeryHard | Identify and select the most significant error in machine-translated sentences. |
| | snarks | VeryHard | Identify sarcastic or ironic statements. |
| | sports_understanding | VeryHard | Answer questions related to sports and select the most appropriate answer. |
| | temporal_sequences | VeryHard | Order events chronologically based on temporal clues. |
| | tracking_five_objects | VeryHard | Track the positions of five objects after a series of shuffling operations. |
| | tracking_seven_objects | VeryHard | Track the positions of seven objects after a series of shuffling operations. |
| | tracking_three_objects | VeryHard | Track the positions of three objects after a series of shuffling operations. |
| | web_of_lies | VeryHard | Evaluate truthfulness in complex scenarios with conflicting statements. |
| | word_sorting | VeryHard | Sort given words alphabetically or by specified criteria. |
| | abstract_algebra | VeryHard | Answer questions about algebraic structures like groups, rings, and fields. |
| | anatomy | Medium | Test knowledge of human anatomical structures and systems. |
| | astronomy | Easy | Answer questions about celestial bodies and astronomical phenomena. |
| | business_ethics | Medium | Evaluate ethical dilemmas in business contexts. |
| | clinical_knowledge | Medium | Test diagnostic and treatment knowledge for medical conditions. |
| | college_biology | Easy | Answer advanced biology questions at university level. |
| | college_chemistry | VeryHard | Solve problems in theoretical and applied chemistry. |
| | college_computer_science | Hard | Test understanding of algorithms and computer systems. |
| | college_mathematics | VeryHard | Solve advanced mathematical problems beyond high school level. |
| | college_medicine | Medium | Answer questions about medical diagnosis and treatment. |
| | college_physics | VeryHard | Solve physics problems requiring university-level knowledge. |
| | computer_security | Medium | Identify security vulnerabilities and protection methods. |
| | conceptual_physics | Easy | Answer physics questions focusing on conceptual understanding. |
| | econometrics | VeryHard | Solve statistical problems applied to economic data. |
| | electrical_engineering | Hard | Test knowledge of circuits, electronics, and power systems. |
| | elementary_mathematics | Hard | Solve basic math problems at the primary school level. |
| | formal_logic | VeryHard | Evaluate the validity of logical arguments and proofs. |
| | global_facts | VeryHard | Answer questions about worldwide demographic and geographic data. |
| | high_school_biology | Easy | Test standard high school biology knowledge. |
| | high_school_chemistry | VeryHard | Answer chemistry questions at secondary school level. |
| | high_school_cs | Easy | Test fundamental programming and computing concepts. |
| | high_school_euro_history | Medium | Answer questions about major events in European history. |
| | high_school_geography | Easy | Test knowledge of countries, capitals, and physical geography. |
| | high_school_gov_politics | Easy | Answer questions about political systems and governance. |
| | high_school_macroeconomics | Medium | Solve problems in national-scale economic systems. |
| | high_school_mathematics | VeryHard | Answer standard high school math questions. |
| | high_school_microeconomics | Easy | Solve problems in individual and firm-level economics. |
| | high_school_physics | VeryHard | Test standard physics knowledge at secondary school level. |
| MMLU | high_school_psychology | Easy | Answer questions about basic psychological concepts. |
| | high_school_statistics | Hard | Solve statistical problems at the high school level. |
| | high_school_us_history | Easy | Test knowledge of key events in American history. |
| | high_school_world_history | Easy | Answer questions about global historical developments. |
| | human_aging | Medium | Test knowledge of biological and social aspects of aging. |
| | human_sexuality | Medium | Answer questions about sexual health and relationships. |
| | international_law | Easy | Evaluate scenarios involving treaties and cross-border legal issues. |
| | jurisprudence | Medium | Answer questions about legal theory and philosophy. |
| | logical_fallacies | Medium | Identify flaws in reasoning and argumentation. |
| | machine_learning | VeryHard | Test understanding of ML algorithms and concepts. |
| | management | Easy | Answer questions about business administration and leadership. |
| | marketing | Easy | Evaluate strategies for product promotion and consumer behavior. |
| | medical_genetics | Easy | Test knowledge of genetic disorders and inheritance patterns. |
| | miscellaneous | Easy | Answer general knowledge questions across diverse topics. |

**Table 12 – Continued from previous page**

| Benchmark | Task | Difficulty | Task Description |
|---|---|---|---|
| | moral_disputes | Medium | Evaluate controversial ethical scenarios. |
| | moral_scenarios | Hard | Judge the appropriateness of actions in ethical dilemmas. |
| | nutrition | Medium | Answer questions about dietary science and health. |
| | philosophy | Medium | Test understanding of major philosophical theories. |
| | prehistory | Easy | Answer questions about human evolution and ancient civilizations. |
| | professional_accounting | Hard | Solve problems in financial reporting and auditing. |
| | professional_law | VeryHard | Answer advanced questions about legal practice. |
| | professional_medicine | Easy | Test clinical decision-making skills for healthcare providers. |
| | professional_psychology | Medium | Evaluate therapeutic approaches and mental health conditions. |
| | public_relations | Hard | Answer questions about media communication strategies. |
| | security_studies | Medium | Test knowledge of international conflict and defense systems. |
| | sociology | Easy | Answer questions about social structures and cultural dynamics. |
| | us_foreign_policy | Easy | Evaluate historical and contemporary American diplomatic strategies. |
| | virology | VeryHard | Test knowledge of viruses and infectious diseases. |
| | world_religions | Medium | Answer questions about belief systems and religious practices. |
| GSM8K | grade_school_math | Hard | Solve grade school level math word problems requiring multi-step reasoning. |
| CNN_DailyMail | text_summarization | Medium | Generate concise and informative summaries of news articles. |

