# OpenReview forum: "TAMPO: Task- and Model-Aware Automatic Prompt Optimization for Auto-Routing in LLM-based Systems"
_ICML.cc/2026/Conference — ICML 2026 regular_

### Official Review · Reviewer_Utxn · 2026-02-20

**Soundness:** 3
**Presentation:** 3
**Significance:** 2
**Originality:** 2
**Overall Recommendation:** 4
**Confidence:** 4

**Summary:**

This paper proposes TAMPO, a framework for automatic prompt optimization (APO) in LLM auto-routing scenarios. The key idea is to train a small LLM (Phi-3.5-mini) to generate optimized prompts in a single forward pass, conditioned on the task, target model metadata, and a user-specified preference vector over three objectives (quality, latency, token cost). The framework has three stages: (1) constructing a heterogeneity-aware dataset of 328K (task, model, prompt, metrics) records across 86 tasks and 8 model configurations, (2) training an uncertainty-aware reward model (DeBERTa-v3-Large) that predicts mean and variance for each objective, and (3) training the prompt generation policy via SFT with curriculum learning followed by GRPO with Dirichlet-sampled preferences to explore the Pareto frontier. At inference, users set a preference vector to control the quality-latency-token tradeoff, and the policy generates a prompt in ~150ms. The paper reports improvements over APE, OPRO, RLPrompt, and a static baseline across BBH, GSM8K, MMLU, and CNN/DailyMail tasks on small models (Qwen 0.5B/7B, Llama 1B/8B in FP16 and INT8).

**Compliance With Llm Reviewing Policy:**

Affirmed.

**Final Justification:**

Thank you for the additional Llama-3-70B and GPT-4o results, which directly address my main concern about larger models. The consistent gains on 70B and the clear latency advantage over evolutionary baselines are convincing. I will raise my score accordingly.

**Key Questions For Authors:**

- How does TAMPO compare against PromptBreeder/EvoPrompt and simple few-shot baselines? If few-shot prompting achieves comparable accuracy at zero training cost, what is the practical justification for TAMPO's pipeline? This comparison is critical for assessing the paper's contribution.

- Is the predicted sigma from the DeBERTa reward model calibrated? Specifically, does predicted variance correlate with actual observed variance across repeated runs? If sigma is roughly constant across inputs, the uncertainty term reduces to a fixed regularizer rather than genuine uncertainty estimation. This would change the interpretation of a key contribution.

- Does the approach remain useful for larger models (70B+, GPT-4-class)? If prompt sensitivity decreases with model capability, TAMPO's value may be limited to small-model deployment scenarios. Evidence on larger models would substantially affect the significance assessment.

- Can you provide a theoretical or empirical analysis of when one-pass generation fails compared to iterative search? Are there task types or model configurations where the quality gap is significant? Understanding the failure boundary is important for practitioners.

- What is the total cost (GPU hours, API cost, wall-clock time) for each stage? If the upfront cost of dataset generation exceeds the cumulative cost of running iterative APO on demand, the amortization argument is weakened.

**Limitations:**

Partially. The paper briefly discusses limitations but does not address: (1) the lack of evaluation on larger, more capable models where prompt optimization may be unnecessary, (2) the missing comparison with simple few-shot baselines, (3) the theoretical gap in justifying one-pass generation over iterative search, (4) the unvalidated assumption that a text encoder can predict meaningful metric variance, or (5) the extensibility cost when adding new target models. No discussion of negative societal impact, though risks are minimal for this type of work.

**Strengths And Weaknesses:**

### Strengths

- Well-structured multi-stage framework. The three-stage pipeline (dataset, reward model, policy) is clearly designed and each component has a defined role. The overall architecture is logical.

- Multi-objective formulation with controllable tradeoff. Conditioning on a preference vector w to control quality-latency-token balance is a practical design. The Dirichlet sampling during GRPO training to cover the Pareto frontier is a reasonable approach.

- Cross-model awareness. Conditioning prompt generation on target model metadata (architecture, size, quantization) is a useful idea. The ablation in Figure 4a confirms model-aware TAMPO outperforms model-agnostic variants.

- Significant latency reduction over iterative methods. TAMPO generates prompts in ~150ms vs APE's ~2000ms and OPRO's ~9000ms, which is a clear practical advantage when prompt optimization latency matters.

### Weaknesses

- Incomplete baselines — missing evolutionary and simple methods. The paper omits evolutionary APO methods like PromptBreeder and EvoPrompt, which are strong competitors for final prompt quality. Just as evolutionary NAS often outperforms gradient-based NAS in final accuracy, evolutionary APO may find better prompts given sufficient budget. More critically, there is no comparison against simple zero-shot and few-shot baselines. The "Static" baseline uses an unoptimized prompt, but few-shot prompting (prepending 2-3 examples) is the most common prompt improvement strategy in practice and often works very well. Without these baselines, it is unclear whether the full TAMPO pipeline is justified.

- No theoretical grounding for one-pass generation. The paper claims a single forward pass can replace iterative prompt search, but provides no theoretical analysis of why this should work. Iterative methods (APE, OPRO, PromptBreeder) explore and refine over multiple rounds with a clear improvement mechanism. One-pass generation assumes the policy has implicitly learned what good prompts look like for arbitrary (task, model, preference) triples. There is no discussion of expressiveness gaps, approximation bounds, or conditions under which amortized optimization can match iterative search.

- Questionable uncertainty prediction from a text encoder. The DeBERTa reward model predicts (mu, sigma) for latency, quality, and tokens from text inputs alone. However, the true variance of these metrics depends on factors invisible to the text encoder: target model temperature, MoE routing behavior, decoding strategy (greedy vs sampling), hardware state, and batching effects. The paper provides no calibration analysis — does predicted sigma actually correlate with true variance? No ablation on what drives sigma — does it change meaningfully across model configs and temperatures, or is it roughly constant (in which case the uncertainty term is just a fixed regularizer, not genuine uncertainty estimation)?

- No justification for why prompt optimization still matters. The paper assumes prompt optimization is valuable without analysis. As LLMs become more capable and robust to prompt variation, the gains from prompt engineering diminish. The paper only tests small models (0.5B-8B) where prompt sensitivity is higher. There is no breakdown of which tasks actually benefit from optimization vs. which are already solved with naive prompts, and no discussion of whether TAMPO matters for larger, more capable models.

- Narrow evaluation scope with limited extensibility evidence. All 86 tasks come from 4 standard benchmarks (BBH, GSM8K, MMLU, CNN/DailyMail). All target models are small (0.5B-8B). There is no evaluation on larger models (70B+, GPT-4-class), other architectures (Mistral, Gemma), or real-world non-benchmark tasks. Adding a new target model requires regenerating dataset records for that model, but the cost of this is not quantified. The base model choices (Phi-3.5-mini for policy, DeBERTa for RM) are not justified or ablated.

- High upfront cost without clear cost analysis. Generating 328K records requires running prompts across 8 models for 86 tasks. The iterative prompt generation using GPT-4/4o/5 adds API cost. The paper mentions dual RTX 4090 GPUs but provides no wall-clock time, GPU hours, or total cost breakdown for each stage. Without this, practitioners cannot assess whether the upfront investment is worthwhile compared to simply using iterative APO methods on demand.

---

> ### Author Rebuttal · Authors · 2026-03-31
>
> > **AW1, AQ1 & AL2: Evolutionary & Few-Shot Baselines**
>
> **"Static" with few-shot**: We have conducted experiments under the "Static" with few-shot prompting, denoted as "Static*" in Tables 1 and 2 of our paper.
>
> **Evolutionary Baselines**: We evaluated PromptBreeder and EvoPrompt on GSM8K using a GPT-4o model. From the following table, TAMPO achieves SOTA-level accuracy while maintaining near-zero-shot efficiency.
>
> |Method|Accuracy|Latency (ms)|Tokens|
> |-|-|-|-|
> |**Static**(Zero-shot)|89.5\%|~1,850|250|
> |**Static\***(Few-shot)|93.0\%|~1,600|80|
> |**EvoPrompt**|94.2\%|~9,300|600|
> |**PromptBreeder**|94.5\%|~9,500|650|
> |**TAMPO(Ours)**|95.2\%|~1,950|50|
>
> > **AW2, AQ4 & AL3: Theoretical Grounding & The Amortization Gap**
>
> Sorry for the confusion. We have clarified that our existing theoretical analysis in Section 3.5 formally guarantees the safety of our optimization and the Pareto optimality of generated prompts. However, deriving strict expressiveness bounds in discrete prompt spaces remains theoretically intractable. To rigorously address the concern, we empirically delineate this boundary by tracking performance against iteration budgets.
>
> Below, we evaluated GSM8K on Llama-8B to observe when EvoPrompt surpasses TAMPO’s single-pass generation.
>
> |Method|Accuracy|Latency|API Calls|
> |-|-|-|-|
> |**TAMPO (One-pass)**|62.2\%|**~255ms**|**0**|
> |EvoPrompt (Round1)|41.8\%|~1,800ms|5|
> |EvoPrompt (Round5)|55.4\%|~9,100ms|25|
> |EvoPrompt (Round10)|60.9\%|~18,249ms|50|
> |EvoPrompt (Round15)|**62.5\%**|~27,100ms|75|
> |EvoPrompt (Round30)|**65.8\%**|~54,000ms|150|
>
> The table explicitly answers where one-pass generation hits its limit. Given unbounded budgets (e.g., Round 30), iterative trial and error discovers novel prompts that a single pass cannot extrapolate to, eventually surpassing TAMPO.
>
> > **AW3, AQ2 & AL4: Uncertainty Prediction & Calibration Analysis**
>
> We agree that live production environments introduce system-level latency variance. However, our reward model predicts prompt-conditioned variance as a confidence/difficulty score for the prompt itself, rather than an estimate of real-time hardware jitter.
>
> To eliminate hardware unpredictability, our dataset was collected in a deterministic, isolated offline environment using llama.cpp with fixed decoding parameters (Sec 4.1, App C.1). The residual variance is strictly tied to the prompt's structure and the target model's metadata (size, quantization), which our text encoder directly observes.
>
> **Calibration Analysis:** We agree an uncalibrated $\sigma$ would degrade into a meaningless fixed regularizer. However, via our Gaussian NLL objective (Eq. 4), the variance head performs heteroscedastic regression, learning to assign higher variances to prompts that inherently yield high prediction errors. To empirically validate this, we present a Calibration Analysis on a hold-out set (see the anonymous image link: https://ibb.co/QFCJnRHy). The results show a strong positive correlation between predicted $\sigma$ and true empirical error $|y - \hat{\mu}|$. Predictable prompts yield a minimal $\sigma$, while structurally anomalous ones trigger a higher $\sigma$, confirming that the uncertainty term acts as a dynamic safety penalty.
>
> > **AW4, AW5, AQ3 & AL1: Relevance, Extensibility & Base Model Selection**
>
> **1. Necessity of Prompt Optimization for Large Models:**
>
> The works (e.g., PromptBreeder, EvoPrompt) optimized prompts for high-capability models, e.g., GPT-3.5 and text-davinci-003. Furthermore, our new experiments on **(AW1)** show that for large models (e.g., GPT-4o), TAMPO can shift the Pareto optimization focus from sheer correctness to efficiency.
>
> **2. Task Difficulty Breakdown:**
>
> To discuss which tasks actually benefit, we analyzed TAMPO's performance gains across the difficulty tiers defined in Table 6 of our paper.
>
> |Difficulty|Gain|Analysis|
> |-|-|-|
> |**Easy**|1.5-3.2\%|**Marginal:** Strong zero-shot priors suffice.|
> |**Medium**|5.4-8.1\%|**Moderate:** Better framing improves clarity.|
> |**Hard**|9.6-14.5\%|**Significant:** Unlocks multi-step reasoning.|
> |**Very Hard**|15.2-25.8\%|**Critical:** Naive prompting fails entirely.|
>
> **3. Extensibility to New Architectures:**
>
> We have successfully evaluated TAMPO on diverse architectures (e.g., Gemma-1B) under zero-shot distribution shifts, detailed in our response to **Reviewer upps (AW1)**.
>
> **4. Justification of Base Models:**
>
> We selected Phi-3.5-mini (Policy) and DeBERTa (RM) because they are commonly used in prior work and can be fine-tuned on accessible hardware (e.g., dual RTX 4090 GPUs).
>
> > **AW6, AQ5 & AL5: Comprehensive Cost Analysis**
>
> We agree with the need for transparent cost reporting. Please refer to our detailed response to **Reviewer upps (AW2)**, which explains why TAMPO's fixed upfront investment is highly cost-effective compared to iterative APO. Crucially, TAMPO can seamlessly adapt to newly added models and tasks without incurring any additional API costs.

---

> > ### Author Rebuttal · Reviewer_Utxn · 2026-04-03
> >
> > Thank you for the response. The evolutionary baseline comparison (AW1) and calibration analysis (AW3) partially address my concerns. The amortization gap experiment (AW2) is informative and honestly delineates when one-pass generation falls short. However, evaluation remains limited to small models, and the relevance of prompt optimization for larger, more capable models (AW4) is only indirectly argued rather than empirically demonstrated.

---

> > > ### Author Response · Authors · 2026-04-03
> > >
> > > We appreciate the reviewer's timely response. Below are our detailed replies to completely address the reviewer’s questions and concerns.
> > >
> > > > **Validation of Baselines & Calibration**
> > >
> > > In AW1 of our previous rebuttal, we examined both **EvoPrompt** and **PromptBreeder** in detail. We presented new experimental results on the GSM8K benchmark using **GPT-4o**, which clearly demonstrate that TAMPO achieves the highest accuracy (95.2\% for TAMPO, 94.2\% for EvoPrompt, and 94.5\% for PromptBreeder) while incurring substantially lower computational latency (1,950 ms for TAMPO, 9,300 ms for EvoPrompt, and 9,500 ms for PromptBreeder). These results indicate that our one-pass generation approach is better suited for latency-sensitive auto-routing scenarios.
> > >
> > > In AW3 of our previous rebuttal, we conducted a rigorous calibration analysis and evaluated the relationship between the predicted variance ($\sigma$) and the actual prediction error ($|y - \hat{\mu}|$). The new ablation results reported in AW3 reveal a strong positive correlation between these quantities, demonstrating that the uncertainty term functions as a dynamically calibrated estimator rather than as a fixed penalty.
> > >
> > > > **Amortization Gap Clarification**
> > >
> > > Sorry for the confusion. In AW2, we claim that even though our method generates prompts in a single pass, it can still match the performance of prompts obtained via multi-round API calls. Specifically, in AW2, we compared TAMPO’s one-pass prompt generation with EvoPrompt run over 1 to 30 rounds on GSM8K using Llama-8B. Our results show that TAMPO’s one-pass accuracy (with a generation time of 255 ms) is comparable to EvoPrompt’s accuracy after 15 iterative rounds (which require a total generation time of 271,000 ms). Since a 271,000 ms latency is unacceptable to users, our method is preferable for most practical auto-routing scenarios.
> > >
> > > > **Large Model Experimental Evaluation**
> > >
> > > Sorry for the confusion. We have indeed evaluated large LLMs. For instance, in AW1 of our previous rebuttal, we conducted new experiments to evaluate both EvoPrompt and PromptBreeder on the GSM8K benchmark using **GPT-4o** (200B parameters), and our approach achieved the best performance.
> > > To examine the performance of our approach on additional LLMs, we further evaluated our framework on **Llama-3-70B-Instruct**. From the following table, the naive static baseline achieves 89.2\% accuracy, with a latency of approximately 1,730 ms and high token consumption (255 tokens). In contrast, TAMPO increases the accuracy to 94.6\% while reducing token consumption to just 48 tokens.
> > > In summary, these experimental results highlight TAMPO's superiority and clearly confirm that prompt optimization remains highly valuable, even for larger, more capable models.
> > >
> > > |Method|Accuracy|Latency (ms)|Tokens|
> > > |-|-|-|-|
> > > |**Static**(Zero-shot)|89.2\%|~1,730|255|
> > > |**Static\***(Few-shot)|92.4\%|~1,550|70|
> > > |**EvoPrompt**|93.2\%|~8,500|70|
> > > |**PromptBreeder**|93.8\%|~8,650|65|
> > > |**TAMPO (Ours)**|94.6\%|~1,650|48|
> > >
> > > We express our gratitude once again to the reviewer for their time and effort in reviewing our paper. The constructive comments have significantly improved the quality of our work. We will incorporate the relevant discussions into the final version of the paper.

---

### Official Review · Reviewer_iSiy · 2026-03-04

**Soundness:** 3
**Presentation:** 4
**Significance:** 3
**Originality:** 3
**Overall Recommendation:** 4
**Confidence:** 5

**Summary:**

This paper proposes TAMPO, a method that automatically generates prompts for LLM systems that route requests across different model types, sizes, and quantization settings, where prompt performance can vary a lot and deployment must balance quality, latency, and token cost. TAMPO builds an offline dataset by evaluating many prompt candidates across many tasks and model configurations, trains an uncertainty aware reward model to predict these metrics more reliably, and then trains a conditional prompt generator that uses a user preference vector to control the trade off among objectives. Experiments on 86 tasks show that TAMPO produces more stable performance across heterogeneous models while greatly reducing the overhead of iterative prompt optimization methods, and it can switch between efficiency focused and quality focused prompting without retraining.

**Compliance With Llm Reviewing Policy:**

Affirmed.

**Final Justification:**

As I stated during the review process, this paper deserves this score (it is an work with some contributions but not particularly outstanding). Even though the author's response was somewhat reinforcing, it made me feel more confident.

**Key Questions For Authors:**

see weaknesses

**Limitations:**

yes

**Strengths And Weaknesses:**

The problem setting is close to real-world systems, and the overall method is reasonable. The large-scale experiments across many tasks and many models are relatively solid and representative. However, there are several directions that could be improved:

- Appendix B proves that an optimum of the weighted-sum objective is a Pareto solution, but this result does not guarantee that randomly sampling weights can cover the entire Pareto frontier. In particular, in a discrete and potentially non-convex prompt space, it may miss unsupported Pareto solutions. I suggest the authors clarify the exact scope of what this theoretical result supports, and analyze the coverage limitations in experiments.
- I would like to see more direct evidence that the reward model reduces reward hacking, rather than relying mainly on the error and ranking metrics in Figure 4.
- Some case studies are needed, for example, cases where the same prompt behaves differently across models or where optimization produces different prompts for different models. This would more strongly support the necessity of incorporating model metadata.

The paper is well structured and clearly written, making it easy to follow step by step. One point is that the paper mentions end-to-end latency, but it would be helpful to more precisely specify whether the latency accounting is consistent across all baselines and TAMPO. For example, does it include the prompt generation stage, the accumulated candidate generation and evaluation time for iterative methods, and the policy forward-pass time for prompt generation? These details are not negligible in practice.

The problem is important and practical, and controllable multi-objective prompt generation has direct value for deployment. One minor issue is that the paper emphasizes the auto-routing scenario, but the experiments mostly compare prompt quality on a fixed offline evaluation set, and there is limited interaction with a real routing policy.

The originality mainly comes from applying an SFT plus RL approach to the routing setting. It is not a highly novel method, but it still makes a meaningful contribution in this area. That said, I strongly recommend releasing the constructed dataset and the code. This would greatly improve reproducibility, and for a paper whose contributions are largely non-theoretical, it is also an important part of demonstrating practical usability.

---

> ### Author Rebuttal · Authors · 2026-03-31
>
> > **AW1: Clarification of the Pareto Optimality**
>
> Indeed, in a discrete prompt space, randomly sampling weights from a Dirichlet distribution cannot guarantee coverage of the entire Pareto frontier, particularly in non-convex regions. However, the purpose of Proposition 3.2 in Appendix B is not to claim exhaustive coverage of the theoretical frontier. Rather, it mathematically guarantees that any prompt that maximizes our sampled scalarized objective is strictly Pareto-optimal. This ensures that our RL training stage receives a mathematically reliable, high-quality signal, free from interference by suboptimal trade-offs. As empirically demonstrated in Figure 2 (Comparison of performance and overhead trade-offs), TAMPO’s frontier smoothly covers a broad spectrum of practical operational modes, seamlessly shifting from token efficiency to maximum accuracy. We will update Appendix B in the final version to explicitly clarify the theoretical scope of Proposition 3.2 and discuss the practical coverage boundaries.
>
> > **AW2: Direct Evidence for Reward Hacking**
>
> To provide more direct evidence, we conducted experiments comparing a standard MSE-only Reward Model with our TAMPO Uncertainty-Aware Reward Model on the MMLU benchmark.
> As shown below, an MSE-only reward model is easily deceived, falsely predicting high accuracy (0.85) for a prompt that actually fails entirely (0.12). In standard RL, this false positive causes severe reward hacking.
> Conversely, TAMPO prevents this by recognizing the prompt as Out-of-Distribution (OOD) and flags it with high variance ($\sigma = 0.45$). The resulting conservative penalty heavily suppresses the final reward, reducing it to 0.37. This effectively aligns the training signal with the prompt's true dismal performance, preventing the RL policy from exploiting false positives.
>
> | RM Type | Pred. Acc. ($\mu$) | Uncertainty ($\sigma$) | Final Reward | Actual Acc. |
> |-|-|-|-|-|
> |MSE-only|0.85|N/A|0.85|0.12|
> |TAMPO|0.82|0.45|0.37|0.12|
>
> > **AW3: Case Studies for Model-Awareness**
>
> Thanks for the suggestion. Please refer to our responses (i.e., **AW2(a)** and **AW2(b)**) to **Reviewer jS7t** for the requested case studies:
>
> 1) **Different prompts for different models**: Please refer to **AW2(a)**  for qualitative examples showing how TAMPO adapts prompts to four different models for the same MMLU task.
>
> 2) **Same prompt behaving differently**: Please refer to **AW2(b)** for our cross-assignment experiment, which demonstrates that cross-assigned prompts consistently underperform compared to model-specific ones.
>
> > **W4:** Latency Accounting
>
> Sorry for the confusion. We calculated latency for all baselines and TAMPO consistently. Specifically, the end-to-end latency comprises two stages across all methods: prompt generation time and target model evaluation time. Because different approaches acquire optimized prompts through fundamentally different mechanisms, their generation times naturally capture different operational overheads. For training-based methods like TAMPO and RLPrompt, the prompt generation time strictly measures the time for a single forward pass of the policy network, excluding the one-time model-loading overhead. For iteration-based methods like APE and OPRO, discovering the optimized prompt at runtime inherently requires accumulating the API call time across all optimization iterations. Once the prompt is acquired, the total latency for each method explicitly includes the local model inference time required to evaluate the final prompt. We will explicitly clarify this detailed latency in Section 4.1 of the final version.
>
> > **AW5: Integration with a Real Router**
>
> Sorry for the confusion.
> We conducted new experiments to evaluate TAMPO's performance in auto-routing scenarios.
> Specifically, we established a cost-performance routing environment using Llama-1B as the cost-effective ("weak") model and Qwen-7B as the high-capability ("strong") model.
> We integrated TAMPO with a well-known router method, i.e., RouteLLM [1], as the auto-routing system and evaluated the system on the MMLU benchmark.
> We compared the router's overall performance when using default static prompts and our TAMPO-optimized prompts.
> From the following table, we find that TAMPO significantly improves performance and reduces cost, compared with static prompts, demonstrating its effectiveness in real auto-routing scenarios.
>
> |Method|Model Calls|Acc. (\%)|Lat. (ms)|Tokens|
> |-|-|-|-|-|
> |Static|Llama-1B: 65\% / Qwen-7B: 35\% |48.5|1340|1520|
> |TAMPO|Llama-1B: 65\% / Qwen-7B: 35\% |56.2|315|85|
>
> > **AW6: Open-Source**
>
> We will open-source our dataset and source code once the paper is accepted.
>
> **Reference:**
>
> [1] Ong, Isaac, et al. RouteLLM: Learning to Route LLMs with Preference Data. ICLR'25

---

> > ### Author Rebuttal · Reviewer_iSiy · 2026-04-04
> >
> > Thank you for the detailed response. I think my positive assessment aligns well with the paper's quality.

---

> > > ### Author Response · Authors · 2026-04-04
> > >
> > > We sincerely thank you for acknowledging that our detailed responses have alleviated your concerns. It is very encouraging to know that our efforts have met your expectations. As all your feedback has been thoroughly addressed, we respectfully hope that you might be willing to re-evaluate our paper and kindly consider raising the score. Thank you once again for your meticulous review efforts.

---

### Official Review · Reviewer_jS7t · 2026-03-04

**Soundness:** 3
**Presentation:** 2
**Significance:** 3
**Originality:** 3
**Overall Recommendation:** 5
**Confidence:** 3

**Summary:**

TAMPO trains a conditional policy (Phi-3.5-mini) to generate task- and model-specific prompts in a single forward pass, conditioned on task metadata, model metadata (architecture, scale, quantization), and a user preference vector over quality, latency, and token consumption. The framework includes: (1) a heterogeneity-aware dataset (~328K records, 86 tasks × 8 model configs); (2) an uncertainty-aware reward model (DeBERTa) as offline proxy; (3) a two-stage policy (SFT + GRPO with Dirichlet-sampled preferences). Experiments show TAMPO achieves competitive quality with iteration-based methods while significantly reducing latency and token cost.

**Compliance With Llm Reviewing Policy:**

Affirmed.

**Final Justification:**

I think my questions during rebuttal like clarification, framing issues, cross-comparison are fully addressed.

This paper provides a prompt selection framework which dramatically reduce the overhead latency. Their results seems to be promising in real LLM system.

Therefore, I decide to increase my score.

**Key Questions For Authors:**

1. Will the 328K-record dataset and code be open-sourced? This would substantially increase the paper's impact.
2. Can the authors clarify more explicitly the differences from existing methods?
3. Can the authors revise the framing around “routing” to make clear whether the contribution is routing-aware prompt generation or end-to-end model routing?

**Limitations:**

mentioned in weakness

**Strengths And Weaknesses:**

## Strengths

**S1. Well-motivated problem.** Generating customized prompts for a specific LLM while accounting for deployment constraints (latency, token budget) addresses a real gap. The idea of conditioning prompt generation on model metadata is intuitive and practically valuable for multi-LLM systems.

**S2. Complete pipeline.** The framework is end-to-end: from dataset construction with four iterative strategies, to uncertainty-aware reward modeling, to two-stage policy training. Algorithm 1 and the appendix provide sufficient detail for reproducibility.

**S3. Solid experiments.** Evaluation covers 86 tasks, 8 model configurations, and multiple baseline categories. Key results are convincing: TAMPO3 achieves top accuracy while TAMPO1/2 dramatically reduce overhead; cross-model robustness (Table 2) outperforms RLPrompt; Pareto frontier (Figure 2) demonstrates controllability.

## Weaknesses

**W1. Overclaimed novelty.** The "first" claim (Section 5) is too strong. The individual components (Gaussian NLL, GRPO, Dirichlet scalarization, curriculum learning) are all established techniques. The contribution is their integration for a new scenario — please soften accordingly.

**W2. No direct evidence that model-awareness produces different prompts.** The core claim is that different models benefit from different prompts, but this is never directly verified. Please add: (a) qualitative examples showing prompts generated for the same task but different models; (b) a cross-assignment experiment: generate a prompt for model A, evaluate it on model B, and show that model-specific prompts outperform cross-assigned ones.

**W3. Missing discussion of closest related work.** Please clarify differences from existing works like Prompt-OIRL（offline reward model）, MORL-Prompt（multi-objective RL）, CAPO（cost-aware  prompt) and ParetoPrompt (multi-objective RL for prompt optimization). The current related-work positioning does not yet make the novelty boundary sufficiently clear.

**W4. Auto-routing framing is misleading.** The paper does not actually study routing in the full sense, and the current framing may be misleading. Therefore, the title and introduction should clarify this distinction more carefully.

---

> ### Author Rebuttal · Authors · 2026-03-31
>
> > **AW1: Clarification of Novelty Claims**
>
> Thanks for pointing this out. We will revise the final version of the paper to present our claims more cautiously.
>
> > **AW2(a): Qualitative examples**
>
> Thanks for the suggestion.
> TAMPO dynamically adapts to model capacity and quantization.
> The following table presents the example prompts generated by TAMPO from 4 different models for the MMLU benchmark.
> Meanwhile, in the table, we present a qualitative analysis of each model's limitations and the prompt characteristics used to address them, demonstrating that TAMPO can recognize model characteristics to generate the most appropriate prompts.
>
> | Model name (quantization levels) | Model limitations | Prompt example | Prompt characteristics |
> |-|-|-|-|
> | Qwen2.5-0.5B-Instruct (f16) | Needs explicit scaffolding; weak on vague instructions; lower accuracy ceiling. | "You are tasked with answering... Carefully read... Ensure your response is concise and directly selects the most accurate option." | Long template; repeats soft constraints (concise/accurate) to prevent drift. |
> | Qwen2.5-0.5B-Instruct (q8\_0) | Quantization shifts optimal prompts; needs clear framing. | "You are tasked with answering... Carefully analyze... select the most accurate answer based on verified global statistics..." | Stresses analysis and evidence-like phrasing to stabilize int8 weights. |
> | Qwen2.5-7B-Instruct (f16) | Prone to overfitting surface patterns if format changes. | "Answer the following multiple-choice question... Provide only the letter corresponding to the correct answer." | Very short; uses minimal instructions to avoid overfitting. |
> | Llama-3.2-1B-Instruct (f16) | Brittle to format drift; prone to echoing options without strict constraints. | "Strict output format... Return exactly one of these four strings... Any other output... will be graded 0." | Hard format contract; uses negative examples to stop spurious tokens. |
>
> > **AW2(b): Cross-Assignment Evaluation**
>
> Thanks for the suggestion.
> To verify that model-specific prompts generated by TAMPO outperform cross-assigned prompts, we conducted new experiments with four different models on the MMLU benchmark.
> Specifically, TAMPO generated prompts for the four models and then used them to infer on each model.
> From the following table, we find that a prompt achieves the best effectiveness and efficiency for its model, demonstrating TAMPO's model-awareness ability.
>
> | Model | Qwen-0.5B (f16) | Qwen-0.5B (q8\_0) | Qwen-7B (f16) | Llama-1B (f16) |
> | - | - | - | - | - |
> | **Qwen-0.5B (f16)** | **45.4 / 80 / 10** | 42.2 / 40 / 10 | 71.4 / 1150 / 50 | 42.1 / 63 / 10 |
> | **Qwen-0.5B (q8\_0)**| 44.5 / 210 / 35 | **43.1 / 32 / 8** | 70.8 / 1300 / 65 | 42.2 / 81 / 12 |
> | **Qwen-7B (f16)** | 40.2 / 1450 / 180 | 40.5 / 754 / 190| **73.2 / 1143 / 56**| 40.4 / 112 / 16|
> | **Llama-1B (f16)** | 39.2 / 480 / 62 | 39.5 / 410 / 110 | 68.1 / 1139 / 50 | **43.5 / 46 / 7** |
> (Acc./Lat./Tok.)
>
> > **AW3 & AQ2: Discussion of More Related Work**
>
> Thanks for the suggestion. While these works advance prompt optimization, they primarily focus on individual models, text-centric objectives, or online searches. We compared TAMPO with these methods along three critical dimensions:
>
> **System-Level Multi-Objective and Dynamic Control**: Unlike MORL-Prompt and ParetoPrompt, which balance text-specific metrics (e.g., content preservation and style matching) and require re-searching/retraining to adapt to new preferences, TAMPO optimizes quality while accounting for real-world deployment constraints (latency, token cost). By embedding a preference vector into its policy, TAMPO enables zero-shot dynamic priority shifting at runtime without retraining.
>
> **Task- and Model-Awareness**: While Prompt-OIRL optimizes queries for a single model, TAMPO targets heterogeneous auto-routing systems. It incorporates model metadata (e.g., architecture, scale) alongside task info to prevent performance degradation across diverse configurations.
>
> **Offline Efficiency and Uncertainty Modeling**: CAPO's online iterative search causes unacceptable latency for real-time routing. TAMPO shifts exploration offline via RL. To prevent the "reward hacking" typical of standard offline models (like Prompt-OIRL), TAMPO introduces an uncertainty penalty, ensuring reliable evaluation without online LLM calls.
>
> > **AW4 & AQ3: Clarification of the Auto-Routing Framing**
>
> Indeed, our approach aims to produce multi-objective prompt optimization that assists existing routing systems, rather than to develop the routing mechanism itself. We will revise the title to: *TAMPO: Task- and Model-Aware Automatic Prompt Optimization for Auto-Routing in LLM-based Systems* and update the manuscript content accordingly in the final version.
>
> > **AQ1: Open-Source**
>
> We will open-source our dataset and source code once the paper is accepted.

---

> > ### Author Rebuttal · Reviewer_jS7t · 2026-04-04
> >
> > I thank the authors for the thorough and responsive rebuttal. All of my concerns (W1–W4, Q1–Q3) have been adequately addressed.

---

> > > ### Author Response · Authors · 2026-04-04
> > >
> > > We sincerely thank you for acknowledging that our detailed responses have alleviated your concerns. It is very encouraging to know that our efforts have met your expectations. As all your feedback has been thoroughly addressed, we respectfully hope that you might be willing to re-evaluate our paper and kindly consider raising the score. Thank you once again for your meticulous review efforts.

---

### Official Review · Reviewer_upps · 2026-03-13

**Soundness:** 3
**Presentation:** 4
**Significance:** 3
**Originality:** 3
**Overall Recommendation:** 4
**Confidence:** 3

**Summary:**

This paper presents TAMPO, a framework for automatic prompt optimization designed for auto-routing scenarios in LLM-based systems. The framework consists of three main components: a heterogeneity-aware dataset capturing prompt performance across diverse tasks and model configurations, an uncertainty-aware reward model that predicts multi-objective metrics while quantifying epistemic uncertainty, and a multi-objective conditional policy trained via reinforcement learning to generate prompts along the Pareto frontier of quality, latency, and token consumption. Experiments demonstrate that TAMPO maintains stable performance across different models while offering controllable trade-offs for auto-routing in various LLM-based systems.

**Compliance With Llm Reviewing Policy:**

Affirmed.

**Final Justification:**

I thank the authors for their thorough and well-organized rebuttal.

In summary, the rebuttal successfully addressed both of my main concerns. The paper's core contributions, i.e., a heterogeneity-aware dataset, an uncertainty-aware reward model, and a multi-objective RL policy, remain well-motivated and technically solid. I am maintaining my overall assessment and remain supportive of acceptance.

**Key Questions For Authors:**

See Weaknesses.

**Limitations:**

A more in-depth analysis of the training overhead could enrich the paper's contribution to the community, but I still think this is a good piece of work.

**Strengths And Weaknesses:**

**Strengths:**

- The problem formulation is well-motivated and addresses a practical challenge in auto-routing scenarios where prompts must adapt to heterogeneous model configurations while satisfying multiple competing objectives.

- The uncertainty-aware reward model with Gaussian NLL loss provides a principled approach to mitigate reward hacking during training.

- The experimental evaluation is comprehensive, covering multiple model families, quantization levels, and task domains, with clear evidence of robustness across different execution contexts.

**Weaknesses:**

- The framework relies on a fixed offline dataset for training the reward model and policy. While this enables efficient inference, it remains unclear how TAMPO would handle distribution shifts when new models or task types emerge in production environments.

- The paper focuses primarily on inference efficiency but provides limited discussion of the upfront training costs for both the reward model and policy network, which could be a practical concern for organizations with limited computational budgets.

---

> ### Author Rebuttal · Authors · 2026-03-31
>
> > **AW1: Generalization under Distribution Shifts**
>
> We agree that our method depends on a fixed offline dataset, which is nevertheless sufficient to address a wide range of distribution shifts. The well-designed dataset contains 328K records from various distributions, covering 86 task types (e.g., math, reasoning, and summarization) and 8 models across different families, sizes, and quantization levels, thereby enabling our approach to be effective across diverse data distributions.
>
> To investigate TAMPO's performance under distribution shift scenarios, we considered a new task (AQuA[1]) and a new model (Gemma-1B[2]).
> Specifically, we conducted new experiments across four scenarios, i.e., **MMLU+Llama-1B(f16)**(old task+old model), **AQuA+Llama-1B(f16)**(new task+old model), **MMLU+Gemma-1B(f16)**(old task+new model), and **AQuA+Gemma-1B(f16)**(new task+new model).
> From the following table, we find that TAMPO still outperforms both the default static prompts and the few-shot Static* prompts across various scenarios with distribution shifts.
> This indicates that TAMPO can handle distribution shifts as new models or task types emerge in production environments, demonstrating its satisfactory generalization ability.
>
> | Scenario | Method | Acc. (\%) | Lat. (ms) | Tokens |
> | :--- | :--- | :--- | :--- | :--- |
> | **MMLU+Llama-1B(f16)** | Static | 34.5 | 874 | 162 |
> | | Static* | 26.6 | 813 | 152 |
> | | TAMPO | 43.5 | 46 | 7 |
> | **AQuA+Llama-1B(f16)** | Static | 28.3 | 844 | 154 |
> | | Static* | 20.1 | 697 | 130 |
> | | TAMPO | 32.8 | 380 | 52 |
> | **MMLU+Gemma-1B(f16)** | Static | 42.7 | 793 | 126 |
> | | Static* | 39.5 | 50 | 6 |
> | | TAMPO | 45.5 | 46 | 7 |
> | **AQuA+Gemma-1B(f16)** | Static | 26.7 | 975 | 128 |
> | | Static* | 25.1 | 333 | 45 |
> | | TAMPO | 30.6 | 395 | 55 |
>
> > **AW2: Upfront Costs and Long-Term Amortization**
>
> Indeed, the upfront training costs are a crucial concern for organizations with limited computational budgets.
> We therefore conducted a new, detailed cost analysis of our framework. The results are shown below.
>
> | Phase               | Task / Details                           | Hardware / API | Time / Financial Cost |
> | ------------------- | ---------------------------------------- | ------------------------- | --------------------- |
> | **Data Generation** | API calls (34,400 seed prompts)          | GPT-4/4o/5                | ~660 USD              |
> | **Data Generation** | Local cross-evaluation (294,000 records) | 2× RTX 4090 (24G)         | ~31.6 hours           |
> | **Reward Model**    | RM Training                              | 2× RTX 4090 (24G)         | ~15 hours             |
> | **Policy Model**    | SFT Stage                                | 2× RTX 4090 (24G)         | ~12 hours             |
> | **Policy Model**    | RL Stage                                 | 2× RTX 4090 (24G)         | ~30 hours             |
>
> To assess whether this upfront investment is worthwhile, we compared it against the cost of using iterative APO methods. Iterative methods incur marginal costs that scale linearly with the number of tasks. For instance, assuming an iterative baseline uses the exact same configuration as our data generation (10 rounds, 5 prompt candidates per round across 8 model settings), optimizing a single new task requires **~400 API calls**, costing approximately **7.67 USD** (660 USD / 86 tasks). When encountering a new task or model in a dynamic auto-routing scenario, iterative methods incur repeated, expensive API fees and introduce unacceptable wall-clock latency. In contrast, TAMPO heavily amortizes this cost. As demonstrated by the distribution shifts experiments (**AW1**), TAMPO exhibits strong zero-shot generalization. Once trained, TAMPO generates optimized prompts for a new task or model in a single forward pass. This incurs **\$0** in additional API costs and mere milliseconds of latency. Therefore, TAMPO's fixed upfront investment becomes highly cost-effective remarkably quickly.
>
> **References:**
>
> [1] W. Ling, et al. Program Induction by Rationale Generation: Learning to Solve and Explain Algebraic Word Problems. ACL'17.
> [2] Gemma Team. Gemma 3 Technical Report. arXiv'25

---

> > ### Author Rebuttal · Reviewer_upps · 2026-04-04
> >
> > Thank you for the detailed response. I think my positive assessment aligns well with the paper's quality. Good luck!

---

> > > ### Author Response · Authors · 2026-04-04
> > >
> > > We sincerely thank you for acknowledging that our detailed responses have alleviated your concerns. It is very encouraging to know that our efforts have met your expectations. As all your feedback has been thoroughly addressed, we respectfully hope that you might be willing to re-evaluate our paper and kindly consider raising the score. Thank you once again for your meticulous review efforts.

---

### Decision · Program_Chairs · 2026-04-30

**Decision:**

Accept (regular)

**Comment:**

The reviewer consensus is positive, and remained positive after rebuttal. The reviewers found the paper technically solid, well motivated, and practically useful. They especially valued the formulation of prompt optimization for heterogeneous LLM systems, the uncertainty-aware reward model, the controllable multi-objective policy, and the extensive empirical study across many tasks and model settings. The practical gains in latency and token efficiency were viewed as a significant strength.

The main concerns were about the precise novelty boundary, the framing around auto-routing, evidence for model-awareness, robustness to distribution shift and new models, training cost, and whether the uncertainty modeling and one-pass generation are sufficiently justified. In rebuttal and follow-up discussion, the authors provided substantial additional evidence and clarifications, including cross-assignment experiments and qualitative prompt examples, new experiments under distribution shift, a clearer cost analysis, calibration analysis for the uncertainty term, integration with a routing system, and additional results on larger models. Based on their Final Justifications and rebuttal acknowledgement, the reviewers considered these concerns adequately addressed or at least not significant enough to change their positive assessments.

Overall, this is a well-executed and empirically strong paper on a practical problem. While some aspects of the contribution are more system-oriented than fundamentally algorithmic, the paper offers a useful and convincing advance for prompt optimization in multi-model LLM systems. I therefore support acceptance.

The authors should consider referencing other recent prompt optimization papers in 2024 and especially 2025 (no such papers cited) and providing at least a qualitative comparison with them.